# Agent-as-a-Judge: Evaluate Agents with Agents

**Mingchen Zhuge** [1 2]  **Changsheng Zhao** [1]  **Dylan R. Ashley** [2]  **Wenyi Wang** [2]
**Dmitrii Khizbullin** [2]  **Yunyang Xiong** [1]  **Zechun Liu** [1]  **Ernie Chang** [1]  **Raghuraman Krishnamoorthi** [1]
**Yuandong Tian** [1]  **Yangyang Shi** [1]  **Vikas Chandra** [1]  **Jürgen Schmidhuber** [2]

## Abstract

Contemporary evaluation techniques are inadequate for agentic systems. These approaches either focus exclusively on final outcomes—ignoring the step-by-step nature of the thinking done by agentic systems—or require excessive manual labour. To address this, we introduce the **Agent-as-a-Judge** framework, wherein agentic systems are used to evaluate agentic systems. This is a natural extension of the LLM-as-a-Judge framework, incorporating agentic features that enable intermediate feedback for the entire task-solving processes for more precise evaluations. We apply the Agent-as-a-Judge framework to the task of code generation. To overcome issues with existing benchmarks and provide a proof-of-concept testbed for Agent-as-a-Judge, we present **DevAI**, a new benchmark of 55 realistic AI code generation tasks. DevAI includes rich manual annotations, like a total of 365 hierarchical solution requirements, which make it particularly suitable for an agentic evaluator. We benchmark three of the top code-generating agentic systems using Agent-as-a-Judge and find that our framework dramatically outperforms LLM-as-a-Judge and is as reliable as our human evaluation baseline. Altogether, we believe that this work represents a concrete step towards enabling vastly more sophisticated agentic systems. To help that, our dataset and the full implementation of Agent-as-a-Judge will be publically available at https://github.com/metauto-ai/agent-as-a-judge

**First four authors made core contributions.** KAUST crafted the dataset. Work done while Mingchen was interning at Meta, with Changsheng leading. [1]Meta AI [2]KAUST. Correspondence to: Mingchen Zhuge <mingchen.zhuge@kaust.edu.sa>, Changsheng Zhao <cszhao@meta.com>.

## 1. Introduction

Recent years have seen multimodal agentic systems move from occasionally being able to solve small toy problems to being regularly deployed for challenging real-world problems (the dream of most AI research). Yet, the current evaluation methods and the available benchmarks for agentic systems are struggling to keep up with these rapid advances, dramatically slowing true progress.

We believe that the current issue with evaluating agentic systems stems from the lack of feedback during the intermediate task-solving stages for these nontraditional systems. Agentic systems think more like a human, often act step-by-step (Wooldridge, 1999) and often hosting very human-like natural language discussions internally to solve problems (Zhuge et al., 2023). And thus agentic systems should be evaluated like a human, with rich evaluative feedback which looks at the full thought and action trajectory; evaluating an agentic system in the traditional way is like evaluating a student using multiple-choice testing—a comparatively unreliable estimator (Park, 2010). For example, while SWE-Bench (Jimenez et al.) is widespread, its evaluation method, which relies solely on the final resolve rate for long-term automated repair tasks, does not effectively pinpoint what is happening within agentic systems that affects the resolve rate. On the other hand, performing a better evaluation with a human is prohibitively expensive. We instead propose that agentic systems should be used to evaluate agentic systems. Inspired by LLM-as-a-Judge (Zheng et al., 2024; Fu et al., 2024; Chen et al.), which uses LLMs to evaluate LLMs, we call this framework Agent-as-a-Judge, of which it is a key extension to the world of agentic systems (see Figure 1). It not only retains the cost-effectiveness of LLM-as-a-Judge but is also equipped with agentic features, allowing it to provide rich intermediate feedback throughout the entire process, as it acts as an agentic system. We apply the Agent-as-a-Judge systems to the problem of evaluating code generating systems—one of the areas where agentic systems have looked the most promising recently.

In code generation, the development of benchmarks has also lagged behind the rapid advancement of agentic systems. HumanEval (Chen et al., 2021), for example, focuses exclusively on algorithmic problems, while MBPP (Austin

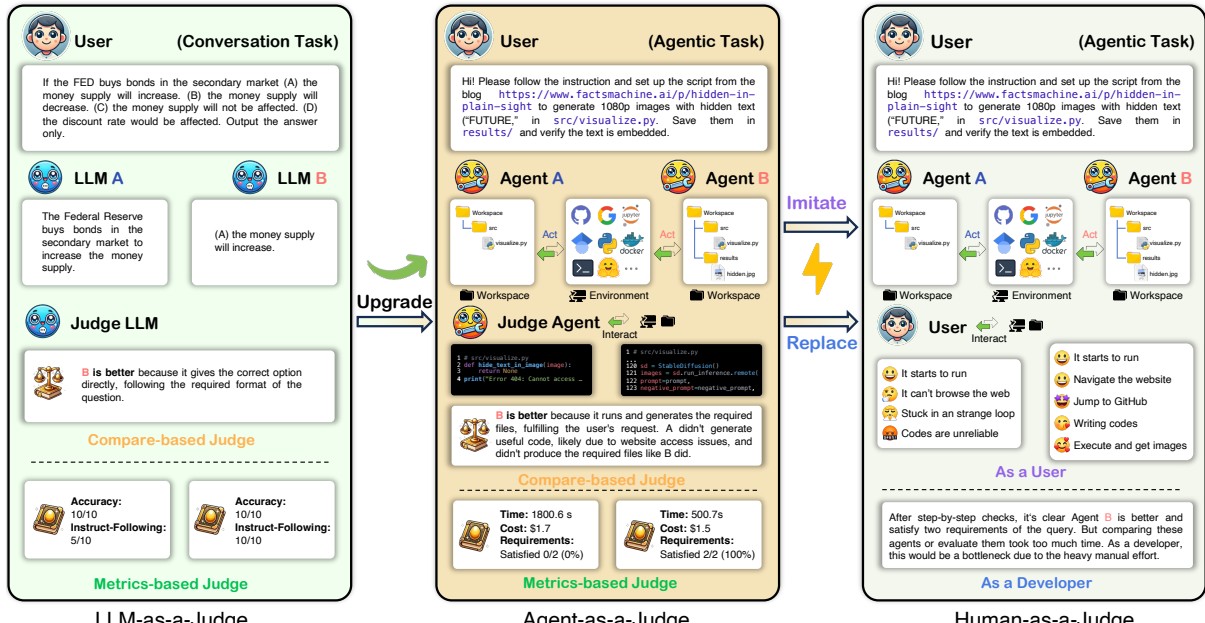

*Figure 1.* In this paper, we introduce the Agent-as-a-Judge framework wherein agentic systems are used to evaluate agentic systems. We compare this to LLM-as-a-Judge, which uses LLMs to evaluate LLMs and for which Agent-as-a-Judge is a natural evolution, and Human-as-a-Judge, where skilled humans labourers manually evaluate an agentic system.

et al., 2021) deals with simple programming tasks. Although they are useful for evaluating the basic skills of foundation models, neither of these two reflects the most practical challenges developers face. As a step away from this, SWE-Bench (Jimenez et al.) did introduce more realistic problems from GitHub, offering a fresh approach to evaluation, but still primarily focuses on automated repairs tasks development process. Concerningly, recent research shows that large language models (LLMs) can already solve over 27% of the tasks in SWE-Bench without needing of advanced agentic systems (Xia et al., 2024). Equally concerning, recent work has begun to introduce mechanisms designed specifically for the individual tasks in the SWE-Bench dataset, leading to a lack of real-world generalization and following Goodhart's law: "When a measure becomes a target, it ceases to be a good measure" (Goodhart, 1976).

To address the aforementioned issues with current code generation benchmarks, we introduce DevAI: the AI Developer Dataset, which contains 55 real-world automated AI development tasks created by expert annotators. We apply three leading open-source code-generating agentic frameworks to the tasks in DevAI: MetaGPT (Hong et al., 2024b), GPT-Pilot (Pythagora.io, 2023), and OpenHands (Wang et al., 2024b). We evaluate their performance using human judges (a painstaking process), LLM-as-a-Judge (Zheng et al., 2024), and our Agent-as-a-Judge framework.

Through human evaluation, we found that GPT-Pilot and OpenHands were each able to satisfy about 29% of the task

requirements in DevAI, but only one full task—showing that DevAI presents a good level of challenge to current systems. When comparing our human judges with our automatic Agent-as-a-Judge framework, we found that Agent-as-a-Judge aligns more closely with the consensus of our human judges (90%) as compared to LLM-as-a-Judge (70%) in all cases tested. In addition, we find that it aligns more closely with this ensemble than the individual human evaluators do, suggesting that—not only is it suitable as a replacement for a human evaluator—but it could in fact be more useful than an average lone human evaluator. In addition, considering the evaluation cost, Agent-as-a-Judge reduces 97.72% of the time and 97.64% of the cost.

In summary, the principal contributions and also the organizations of our development steps are:

• (**Step 1** - Section 2) We release the DevAI dataset, which consists of 55 comprehensive AI development tasks with accompanying tags, individual hierarchical requirements, and individual preferences.

• (**Step 2** - Section 3) We benchmark three top open-source code generation agentic frameworks in DevAI, providing a more comprehensive analysis than previous evaluations of them.

• (**Step 3** - Section 4) We introduce the general Agent-as-a-Judge concept, allowing agentic systems a fair and rich evaluation without the traditional cost this would require in human labour.

• (**Step 4** - Section 4) We demonstrate that an Agent-as-a-Judge outperforms an LLM-as-a-Judge and performs comparably to human evaluators in our proof-of-concept.

## 2. (Step 1) Crafting a Benchmark for Automated AI Development

### 2.1. Motivation

**Background** The code generation domain is an area where agentic systems have seen significant industrial deployment during the past 2 years (e.g., see Devin[1] and Cursor[2]). However, in code generation, there isn't yet a benchmark that accurately reflects realistic user queries for developing complete AI systems. We believe this is because of the difficulty to evaluate such complex, real-world tasks. For example, while many companies advertise their systems based on the performance of it on benchmarks like SWE-Bench (Jimenez et al.) (for automated repair) or HumanEval (Chen et al., 2021) (for algorithmic tasks), these benchmarks cover only a small bit of an actual development process. A benchmark that can evaluate the entire development process—ideally in a way that can help understand the degree to which current AI methods can reduce human labour—is missing.

**Topic** We chose automated AI development as our main topic. While AI and ML tasks are often more complex, they follow clear, standard procedures. For example, data processing typically comes first in an AI pipeline, and performance reporting goes at the end. We believe this topological nature can help better monitor the development process and provide useful signals to the agentic systems.

**Goals** An ideal benchmark should address critical issues in automated development by focusing on three key factors. First, it should reflect practical software scenarios, where tasks are often too complex for a single LLM, requiring human or agentic systems. Second, it should emphasize the development process, not just final outcomes (e.g., `pass@1` rates offer limited feedback and fail to highlight intermediate problems). Lastly, the evaluation should be computationally cost-effective and efficient, avoiding long training times or excessive manual oversight.

### 2.2. The DevAI Dataset

Motivated by the ideas outlined above, we propose the DevAI dataset. DevAI consists of a carefully curated set of 55 tasks, each defined by **(1)** a plain text query that describes an AI development task; **(2)** a set of plain text requirements (for a total of 365 requirements), each with a set of dependencies connecting them to other requirements; and **(3)** a set of preferences (for a total of 125 preferences) which represent

softer requirements. DevAI is structured so that an agentic system starts by receiving a user query to begin development. The system is then evaluated on how well it meets the requirements, with preferences serving as optional, softer criteria. An example of one of the DevAI tasks can be seen in Figure 7 and the full suite is available at [REDACTED]

The tasks in DevAI are relatively small-scale but cover commonly used key development techniques. As shown in Figure 2, our tasks are tagged and cover a variety of key areas in AI: supervised learning, reinforcement learning, computer vision, natural language processing, generative models, and others. Each of the tasks is a real-world problem that could be given to a research engineer, while simultaneously being relatively inexpensive computationally to run so as to reduce the cost of evaluating a method on this benchmark. Details of the sample collection and human labeling process for DevAI are provided in Appendix E.

The requirements belonging to each task represent a milestone in the comprehensive development process and are arranged as a directed acyclic graph (similar to the work by He et al. (2021)), with requirements such as visualizing results depending on correct data loading and modeling. This allows for more comprehensive non-sparse feedback than a binary success metric. Furthermore, the inclusion of milestones makes simple memorization not viable as a solution strategy as code that performs the entire task is unlikely to exist online at this time.

### 2.3. Preliminary Benchmark

We first conduct experiments to collect development outcomes from different frameworks, which serve as baselines in the DevAI dataset. We test three of the most popular open-source frameworks (which we will refer to as "**AI developers**"): MetaGPT (Hong et al., 2024b), GPT-Pilot (Pythagora.io, 2023), and OpenHands (Wang et al., 2024b)—all selected for their strong community acceptance (each having over 30,000 stars on GitHub).

**Experiment Setup** All of these three systems require a language model as a back-end engine, for which we use `gpt-4o-2024-05-13`, a state-of-the-art language model. These AI developers were given a time-limit of 1800 seconds to solve each task and were forcefully halted if they exceeded this time limit (we imposed this constraint, which was visible to the AI developers, as detailed in Appendix I). We capture the outputs generated during the automated development process, including code, files, and other artifacts. Additionally, we record key decisions and actions made by the agentic systems through some custom instrumentation code, resulting in a thought trajectory for each of the agentic systems.

**Analysis** The basic statistics are shown in Table 1.

---

[1] https://www.cognition.ai/blog/introducing-devin
[2] https://www.cursor.com/

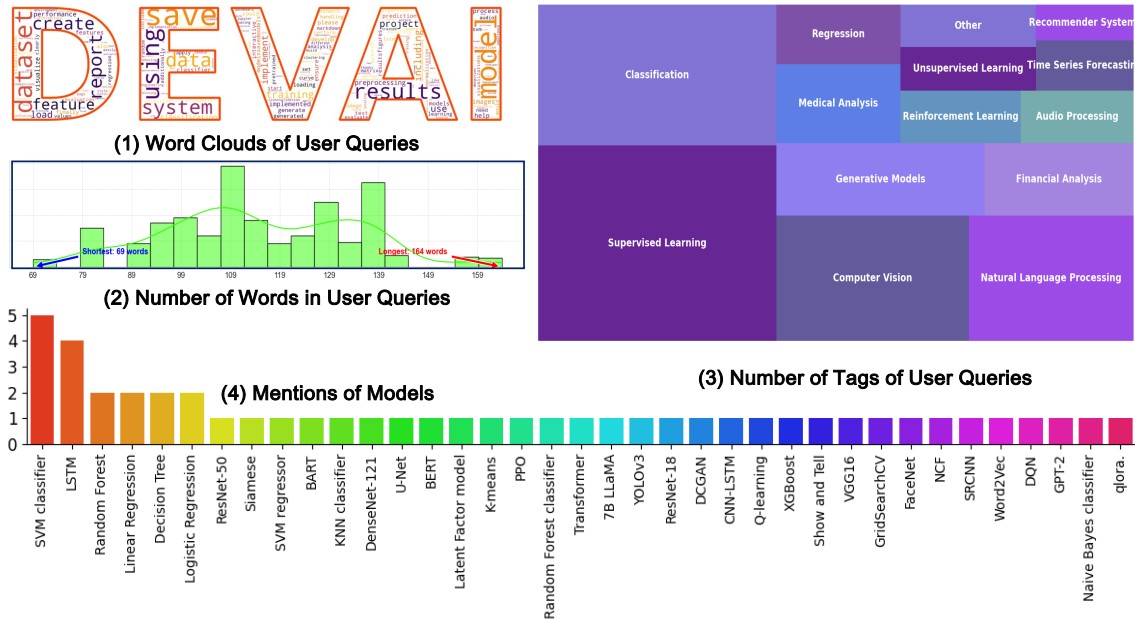

*Figure 2.* **Distribution of DevAI Tasks** (1) DevAI focuses on AI development tasks and so terms such as "dataset," "model," and "results" are particularly common in the queries. (2) The first 53 tasks in DevAI all have a one-paragraph query but of varying lengths (note that task 54 and 55 are excluded here as they are outliers, representing the longest and most complex tasks in the dataset). (3) Each task has one or more tags. The prevalence of supervised learning here reflects the fact that it dominates many machine learning applications. (4) SVM classifiers (Cortes, 1995) and LSTM models (Hochreiter, 1997) are two of the most widely used architectures—a fact reflected by DevAI.

MetaGPT is the most cost-efficient (1.19 USD), while Open-Hands is the most expensive (6.38 USD). In terms of development time, OpenHands completes tasks in an average of 362.41s, while GPT-Pilot takes the longest at 1622.38s. On average, a full evaluation on DevAI with one of these three took around 210.65 USD and 14 hours to perform. While running, GPT-Pilot generates the most output tokens at 59707 tokens, whereas OpenHands processed the most at 1252482 tokens while producing the fewest at 8457 tokens. This suggests that OpenHands's internal communication is more complicated but is more parsimonious in its decisions.

MetaGPT, while being the most cost-effective, generates fewer saved code files (0.42), suggesting it may be less inclined to save files. In contrast, GPT-Pilot generates the most saved files (3.84), reflecting a more prolific output. The difference in saved code lines, with GPT-Pilot saving 273.33 lines versus MetaGPT's 11.15, underscores GPT-Pilot's extensive output. Meanwhile, OpenHands, despite handling larger inputs, seems less focused on executing code to generate files, as evidenced by its lower file output (2.53 saved files). These statistics align with real user experiences (as discussed in Appendix F).

**Evaluations** Note that the results in Table 1 are not directly indicative of performance, but provide valuable intuition about the practical utility of DevAI. The generated workspaces (generated files, code, etc.) and trajectories are utilized in subsequent experiments to perform evaluations using Human-as-a-Judge (section 3), LLM-as-a-Judge, and Agent-as-a-Judge (section 4).

## 3. (Step 2) Manual Evaluation on DevAI (Human-as-a-Judge)

To determine the pragmatic validity of DevAI and to accurately estimate the actual code-generating abilities of current state-of-the-art agentic systems, in this section, we run and then manually evaluate the application of three AI developer baselines to DevAI. In Section 4, we show how this evaluation can be automated.

### 3.1. Benchmark Baselines by Human-as-a-Judge

**Human Evaluation Setup** After obtaining the baseline executions and conducting basic statistical analysis, we have three expert human evaluators (referred to here by their anonymous names: `231a`, `38bb`, and `cn90`) review the outputs of AI developer baselines to assess whether each requirement was satisfied. We have two rounds of human evaluations. To capture the bias inherent in typical human evaluation (this is desirable to capture here as it represents a likely scenario in deployment), in the first round, our evaluators first discussed the basic standards but were given minimal instructions. The templates the evaluators were given for the evaluation and their self-reported post-hoc descriptions of how they resolved ambiguities are reported

*Table 1.* **Preliminary Statistics of AI Developers.** We compare three leading open-source code agents using metrics such as average cost, average time, and the average number of generated files.

| Metric | MetaGPT (Hong et al., 2024b) | GPT-Pilot (Pythagora.io, 2023) | OpenHands (Wang et al., 2024b) |
|---|---|---|---|
| 📝 **Basic Statistics** | | | |
| Version | Data Interpreter (Hong et al., 2024a) | 0.2.13 | CodeAct v1.9 (Wang et al.) |
| (1) Average Cost | $1.19 | $3.92 | $6.38 |
| (2) Average Time | 775.29s | 1622.38s | 362.41s |
| (3) Average Input Tokens | 152863 | 606707 | 1252482 |
| (4) Average Output Tokens | 28546 | 59707 | 8457 |
| (4) Average Saved Code Files | 0.42 | 3.84 | 2.53 |
| (5) Average Saved Code Lines | 11.15 | 273.33 | 96.56 |
| (6) Average Saved Files | 4.42 | 5.91 | 3.60 |

*Table 2.* **Human-as-a-Judge for AI Developers.** (I) and (D) represent independent performance versus performance considering task dependencies. 👥 indicates multiple experts evolved, and ⬜ means the evaluations use white-box testing (allowing access to the generated workspace, human-collected trajectories, and open-source codebases). The results were derived from expert judgments and deliberations (see Appendix H).

| Metric | MetaGPT (Hong et al., 2024b) | GPT-Pilot (Pythagora.io, 2023) | OpenHands (Wang et al., 2024b) |
|---|---|---|---|
| 👥 / ⬜ **Human-as-a-Judge** | | | |
| (A) Requirements Met (I) | 22.13% | 44.80% | 42.89% |
| (B) Requirements Met (D) | 6.55% | 28.96% | 28.68% |
| (C) Self-Termination | 41.81% | 5.45% | 54.54% |
| (D) Task Solve Rate | 0.00% | 1.81% | 1.81% |

in Figure 11 in Appendix H.

After the initial round of human evaluations (which totaled an estimated total of 58 human hours), we asked our evaluators to discuss and reach a consensus on their assessments (which took an estimated total of 28.5 additional human hours). This consensus, achieved after long sessions of debate, was used as the final human evaluation result for each method.

**Performance Analysis** The results of this experiment are shown in Table 2. We found that the two best-performing methods (GPT-Pilot and OpenHands) could satisfy about 29% of the requirements (or around 44% if prerequisites are ignored) but only on one task could they meet all the requirements. This highlights that DevAI offers a considerable but appropriate level of challenge for current and future methods. Moreover, the fulfillment of intermediate requirements aligns with our expectations (see Section 2) that DevAI provides richer feedback by uncovering how agentic systems falter during the process instead of just focusing on a single performance metric at the end.

### 3.2. Judging Human-as-a-Judge

**Disagreement Analysis** To analyze the presence of inductive bias and the reliability of the Human-as-a-Judge paradigm here, we calculate the disagreement rate between individual evaluators (shown in Figure 3). The results indicate that the disagreement rates between pairs of evaluators range from around 10% to 30%. Although each human

evaluator has over five years of experience in AI research and development, the disagreement highlights the inherent challenges of evaluating AI development.

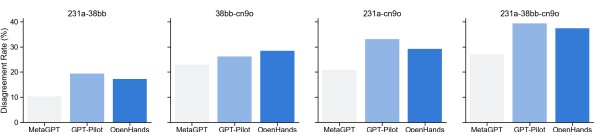

*Figure 3.* Between the three human evaluators, a large amount of disagreement was observed in their individual evaluations—highlighting the inherent unreliability of human evaluation.

Due to the complexity of a complete AI development task, which typically involves multiple steps with varying outcomes at each step, humans can easily make errors when critical information is missed, such as environment feedback indicating small but severe coding errors or bugs. Additionally, some disagreements are not necessarily incorrect but arise from differing perspectives on how ambiguity should be resolved.

To determine if the disagreement between the three human judges is too large for them to serve as a strong baseline, we recruited ten additional experts and had them evaluate a random selection of 7 task samples. We observed that the majority vote of these additional experts had a 95.23% agreement rate with the consensus result of our other experts on these 7 tasks and a 97.67% agreement rate with the majority vote. See Appendix P.

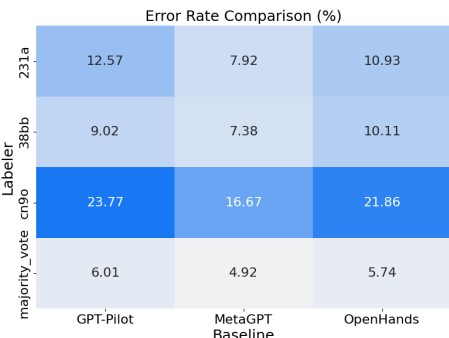

*Figure 4.* Mismatch between the individual evaluations and the consensus evaluation. Note that the majority vote showed the smallest deviation from the consensus evaluation.

**Error Analysis** As previously noted, the evaluators engaged in a round of debating after their initial evaluations until they reached a consensus on each requirement in each task (with the results of this consensus evaluation shown in Table 2).

In our Human-as-a-Judge pipeline, evaluators could be convinced by evidence from others and acknowledge their judgment errors, adjusting their answers accordingly. This can be used to approximate individual errors. If the consensus evaluation more accurately predicts any extant ground truth, we would expect the majority vote from the individual evaluations to more closely approximate this than any single evaluation, due to the fundamental properties of ensemble classifiers (see Hastie et al. (2009)).

While the consensus evaluation may not represent the absolute ground truth (we acknowledge that some quantity of error likely would still exist even after this procedure), we expect the consensus evaluation to more accurately approximate the extant ground truth (Clemen, 1989). If this holds, the majority vote should align more closely with the consensus than with any individual evaluation. As shown in Figure 4, this is indeed the case.

As seen in the results, although significant errors occur among all evaluators, the majority vote effectively corrects most of these errors. Notably, `cn9o` made the most errors (for example, 23.77% in evaluating GPT-Pilot). After applying the majority vote from all three evaluators, the overall error rate dropped to 6.01%, demonstrating the inherent benefits of majority voting.

**Conclusion** Human judgment errors are inevitable. To reduce them, we suggest two methods. First, like in this work, introduce a debate round after each judgment, where individuals present evidence and either persuade others or adjust their own opinions after discussion. This is particularly important when there are only a few evaluators, as majority voting with a small group can still lead to errors (around 5% compared to consensus evaluation, as shown

in Figure 4). The second approach involves assembling a larger panel of experts (more is better when their accuracy exceeds 50% (Grofman et al., 1983)), with over 5 people recommended by Hastie & Kameda (2005); Larrick & Soll (2006), and relying on a majority vote. However, due to the high cost of engaging more experts and the fact that this is not always feasible in practice, we argue for the former.

## 4. (Step 3&4) Evaluating Agents with Agents (Agent-as-a-Judge)

Human evaluation, while somewhat reliable, is time-consuming and requires significant expertise. To address this, we propose the Agent-as-a-Judge framework. If such an agentic system could evaluate like a human, it would reduce the need for human involvement, eliminating the trade-off between evaluation thoroughness and effort.

### 4.1. Spefic Agent-as-a-Judge for Code Generation

Based on our prior experiences with agent design and by imitating the human evaluation process, we initially designed eight modular, interacting components that form the foundation of our tailor-made Agent-as-a-Judge.

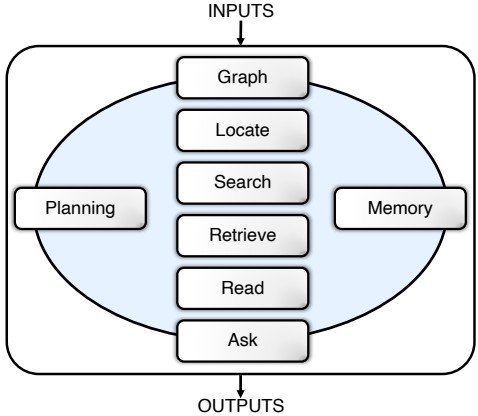

*Figure 5.* Initial diagram of Agent-as-a-Judge.

**(1) The graph module** constructs a graph that captures the entire structure of the project, including files, modules, and dependencies. It can also break down chunks of code into code snippets. **(2) The locate module** identifies the specific folder or file referred to by a requirement. **(3) The read module** goes beyond simple file parsing, supporting the reading and understanding of multimodal data across 33 different formats, including code, images, videos and documents. This allows the agent to cross-reference various data streams and check different kinds of requirement. **(4) The search module** offers a contextual understanding of code and can quickly retrieve highly relevant code snippets, as well as the nuances behind them (e.g., hidden dependencies). **(5) The retrieve module** extracts information

*Table 3.* **AI Judges and Their Shift/Alignment with Human-as-a-Judge.** We compare the results of LLM-as-a-Judge and Agent-as-a-Judge with Human-as-a-Judge. (I) represents performance on independent tasks, while (D) represents performance considering task dependencies. **Note:** ▢ gray-box settings use carefully manually collected trajectory data (which is nearly inaccessible in practical situations, see Appendix J). In contrast, ■ black-box setting doesn't need to access to such data. The red scores represent the absolute judge shift compared with Human-as-a-Judge (*e.g.*, 2.74%).

| Metric | MetaGPT (Hong et al., 2024b) | GPT-Pilot (Pythagora.io, 2023) | OpenHands (Wang et al., 2024b) |
|---|---|---|---|
| **■ LLM-as-a-Judge** | | | |
| (a) Requirements Met (I) | 19.39% (2.74%) | 12.56% (32.24%) | 11.47% (31.42%) |
| (b) Requirements Met (D) | 1.63% (4.92%) | 4.09% (24.87%) | 2.18% (26.50%) |
| (c) Task Solve Rate | 0.0% (0.0%) | 0.0% (1.81%) | 0.0% (1.81%) |
| **Alignment Rate ↑** | 84.15% | 65.30% | 60.38% |
| **■ Agent-as-a-Judge** | | | |
| (I) Requirements Met (I) | 25.40% (3.26%) | 53.00% (8.20%) | 42.62% (0.27%) |
| (II) Requirements Met (D) | 5.73% (0.81%) | 39.89% (10.93%) | 26.50% (2.17%) |
| (III) Task Solve Rate | 0.0% (0.0%) | 5.45% (3.64%) | 1.81% (0.00%) |
| **Alignment Rate ↑** | 88.52% | 83.88% | 90.44% |
| **▢ LLM-as-a-Judge** | | | |
| (a) Requirements Met (I) | 28.68% (6.55%) | 38.79% (4.10%) | 43.16% (0.27%) |
| (b) Requirements Met (D) | 17.75% (11.20%) | 33.06% (4.10%) | 32.24% (3.56%) |
| (c) Task Solve Rate | 1.81% (1.81%) | 3.63% (1.82%) | 7.27% (5.46%) |
| **Alignment Rate ↑** | 68.86% | 71.85% | 70.76% |
| **▢ Agent-as-a-Judge** | | | |
| (I) Requirements Met (I) | 23.49% (1.35%) | 46.44% (1.64%) | 43.44% (0.54%) |
| (II) Requirements Met (D) | 6.01% (0.54%) | 30.60% (1.64%) | 28.14% (0.53%) |
| (III) Task Solve Rate | 0.0% (0.00%) | 5.45% (3.64%) | 3.63% (1.82%) |
| **Alignment Rate ↑** | 92.07% | 86.61% | 90.16% |
| **👤 / ▢ Human-as-a-Judge** | | | |
| **Alignment Rate** (38bb) | 92.63% | 90.98% | 89.89% |
| **Alignment Rate** (cn9o) | 83.33% | 76.23% | 78.15% |
| **Alignment Rate** (231a) | 92.07% | 87.43% | 89.07% |
| Average of individuals | 89.34% | 84.88% | 85.70% |
| **Alignment Rate** (Majority Vote) | 95.08% | 93.98% | 94.26% |

from long texts, identifying relevant segments in trajectories. With context from the above, **(6) the ask module** determines whether a given requirement is satisfied.**(7) The memory module** stores historical judgment information, allowing the agent to build on past evaluations. Finally, **(8) the planning module** plans out actions, allowing the agent to strategize and sequence tasks based on the current state and project goals.

Our initial design of the Agent-as-a-Judge is shown in Figure 5. After conducting comprehensive ablation studies, we found that the modular combination of **(1)**, **(2)**, **(3)**, **(5)**, and **(6)** achieved the highest performance (see Appendix C). A sample of the dynamic evidence collected by the Agent-as-a-Judge is shown in Appendix M. We hypothesize this is because Agent-as-a-Judge needs high-quality factual information and is sensitive to noise. For example, while our design of the planning module introduces promising decision-making for future actions, the procedure is unstable. Initially, we hoped that historical information from the memory module would help to assess current requirements. However, it proved detrimental, as any errors in previous judgments could lead to a chain of errors, negatively affecting current decisions. Besides, the current workspaces

generated by developer agents, having only hundreds of lines of code, cannot fully benefit from the search module. The details of these findings are explained in Appendix K. Note that a perfect Agent-as-a-Judge is not the focus of this proof of concept, and thus, we leave the utilization of advanced agentic optimization methods for Agent-as-a-Judge, such as automated prompt optimization and workflow design (Zhuge et al.; Hu et al., 2024), for future work.

### 4.2. Judging Agent-as-a-Judge and LLM-as-a-Judge

**Judge Shift** Judge Shift measures deviation from the Human-as-a-Judge consensus results, with lower values indicating a closer alignment. As shown in table 3, Agent-as-a-Judge consistently outperforms LLM-as-a-Judge across tasks, particularly those with task dependencies. For example, in Requirement (I), Agent-as-a-Judge shows a Judge Shift as low as 0.27%, while LLM-as-a-Judge reaches 31.24% for OpenHands. This underscores Agent-as-a-Judge's stability and suitability for meeting task requirements. Furthermore, in the gray-box setting, both Agent-as-a-Judge and LLM-as-a-Judge show even better results than their performance in the black-box setting.

**Alignment Rate** The Alignment Rate reflects how closely the AI Judges' evaluations align with human consensus across all 365 requirements. It is defined as the percentage of requirement evaluations that are the same as the Human-as-a-Judge consensus evaluation. Compared to LLM-as-a-Judge, Agent-as-a-Judge consistently achieves a higher Alignment Rate, closely matching human judgments. For example, when evaluating OpenHands, Agent-as-a-Judge reaches 92.07% and 90.44%, surpassing LLM-as-a-Judge's 70.76% and 60.38% in both gray-box and black-box settings. This suggests that Agent-as-a-Judge is more accurate and human-aligned.

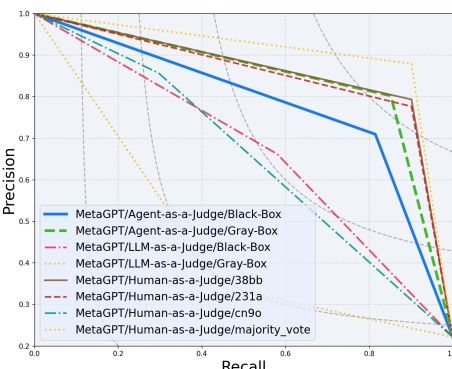

*Figure 6.* PR Curves comparing judge Methods.

**PR Curves** Judging developer agents is a class-imbalanced task, where meeting requirements is much rarer than failing. Metrics like judge shift and alignment rate can be misleading. For example, since MetaGPT rarely meets requirements, LLM-as-a-Judge easily identifies most cases as negative (achieving 84.15% in the black-box setting). PR Curves offer a clearer performance measure by balancing precision and recall. This shows that, in some cases, Agent-as-a-Judge can nearly replace human evaluators. Our observations indicate the relative reliability of evaluation methods as: LLM-as-a-Judge < Single-Human-as-a-Judge < Agent-as-a-Judge < Ensemble of Human Judges. Future advancements in foundation models and Agent-as-a-Judge designs may shift this order.

*Table 4.* **Component Ablation Studies for Agent-as-a-Judge.** We analyze the impact of adding various components (ask, graph, read, locate, and retrieve) on the performance of Agent-as-a-Judge for judging OpenHands.

| Metric | + ask | + graph | + read | + locate | + retrieve |
|---|---|---|---|---|---|
| **Agent-as-a-Judge Performance** | | | | | |
| Alignment Rate | 65.03% | 75.95% | 82.24% | 90.44% | 90.16% |

### 4.3. Ablations For Agent-as-a-Judge

We conduct ablations to evaluate the impact of adding different components on Agent-as-a-Judge's performance.

The components analyzed include ask, graph, read, locate, and retrieve. The component ablation study for Agent-as-a-Judge reveals key insights into the performance gains from adding specific functionalities. With only ask component, the agent achieves a 65.03% alignment rate. Adding the graph component increases performance to 75.95%, as the agent can better understand relationships between files. The introduction of read further improves the alignment rate to 82.24%, reflecting the value of direct access to the contents of the file. Incorporating locate brings a substantial boost to 90.44%, as the agent can efficiently target files relevant to the requirements. Adding retrieve does not always provide a significant benefit in this case. We found the retrieve module effective for judging MetaGPT and GPT-Pilot, as it provides valuable trajectory information (as shown in Table 3). However, it is less effective for OpenHands, which sometimes fails to execute files, resulting in missing responses. In such cases, judgment without trajectories remains viable.

### 4.4. Cost Analysis

Our three evaluators a self-reported total of 86.5 hours. With a 15 USD hourly wage (assuming this would buy a subject expert in AI), a full evaluation under DevAI would cost around 1297.50 USD. In comparison, Agent-as-a-Judge cost only 30.58 USD (2.29%) in API calls and took only 118.43 minutes (2.36%). LLM-as-a-Judge was faster at 10.99 minutes, but due to the absence of intelligent context selection by the Agent-as-a-Judge's modules, it still cost 29.63 USD.

## 5. Related Work

Agentic systems is a highly active research area, so we only detail the most closely related works here. We provide a treatment of the marginally less relevant related works in Appendix D.

**AI Developers** AI in software development is growing fast (Liu et al., 2024). AI-driven developers have been applied to directly imitate software companies (Hong et al., 2024b; Qian et al., 2024a), debug code (Yang et al., 2024a), run data science methods (Guo et al.; Hong et al., 2024a; Li et al., 2024; Qiao et al., 2023), and even write academic papers (Lu et al., 2024a).

**Benchmarks for AI developments** Benchmarks like MLAgentBench (Huang et al., 2024), ML-Bench (Liu et al., 2023), and SUPER (Bogin et al., 2024) all focus on benchmarking agentic systems using AI tasks. However, DevAI distinguishes itself from all of these by focusing on realistic user queries that target a complete development cycle. It further includes a more comprehensive evaluation with multiple hierarchical requirements and preferences for each task.

Comparatively, MLAgentBench (Huang et al., 2024), for example, focuses on final performance for a limited set of well-known tasks, which risks overfitting and fails to assess a system's generalization or adaptability.

**AI Judges**   Several works have looked at using AI systems as judges. The work by Chan et al.; Zhao et al. (2024), for example, extends LLM-as-a-Judge to have multiple LLMs in their evaluation process for conversational tasks. Unlike Agent-as-a-Judge, they employ a trivial agentic system and apply it only to evaluate LLMs under traditional evaluation setups. In contrast, (Lu et al., 2024b) uses a single LLM-based evaluator but, unlike LLM-as-a-Judge, applies this to multimodal tasks rather than just for evaluating LLMs. Less relevant are frameworks like those by Chen et al. (2024b); Arora et al. (2024); Mündler et al. (2024), where intermediate signals are used during coding development.

## 6. Discussion and Conclusion

**Discussion**   A key power of the Agent-as-a-Judge that we have not exploited here is the feedback it provides being directly used by the agentic system being evaluated. Perhaps the greatest strength of the Agent-as-a-Judge framework is that an agentic system could use it to fix issues in their solutions to complex multistage problems on the fly—something older delayed feedback methods did not permit. In addition, a cycle of mutual improvement between Agent-as-a-Judge and the evaluated agents, where both evolve together through iterative feedback, presents a promising perspective.

In this work, we focus on code tasks, however, the concept of "Agent-as-a-Judge" is inherently general. In our experience, transforming our current implementation to other domains will not necessitate overcoming significant bottlenecks. This is because our implementation is modular, comprising components such as `graph`, `read`, `locate`, `ask`, and `retrieve`. By modifying the prompts and workflow, it should not be difficult to transfer the system to other domains—a common practice in agentic projects. Our `read` module already supports multimodal data (code, text, images, etc.), and the `ask` module can be adapted to verify arbitrary requirement criteria. However, we still encourage the reader to design domain-specific "Agent-as-a-Judge" systems for more accurate usages.

**Conclusion**   In this work, we introduced the Agent-as-a-Judge method to use agentic systems to evaluate agentic systems. We simultaneously released DevAI: a new benchmark that evaluates the code-generating ability of agentic systems on complete AI development tasks when used with Agent-as-a-Judge. We went on to show that Agent-as-a-Judge outperforms existing methods on this task and that it performs similarly to an ensemble of expert human evalua-

tors. Altogether, we believe that the above opens the door for scaling up agentic far more than before.

## Acknowledgements

This publication was supported by funding from KAUST Center of Excellence on GenAI under award number 5940, as well as, the SDAIA-KAUST Center of Excellence in Data Science and Artificial Intelligence.

## Impact Statement

This work introduces an automated, agent-driven evaluation framework designed to improve the scalability, efficiency, and transparency of assessing autonomous systems. By substantially lowering the time and cost required for reliable review, it democratizes access to rigorous evaluation, enabling research groups and industry practitioners with limited resources to iterate more quickly and safely. However, automating judgment also carries risks: if the judge-agent's own reasoning is flawed or biased, errors may propagate unchecked; over-reliance on automated assessment could reduce critical human oversight, potentially missing emergent ethical or safety concerns. In the broader context, Agent-as-a-Judge can accelerate the development of robust multi-agent systems, while fostering a culture of transparent, reproducible evaluation that is essential for the responsible advancement of machine learning.

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

# A. Task Sample

---

**Task 51: Devin_AI_Software_Engineer_Plants_Secret_Messages_in_Images**

### Query

*Hi! Please follow the instructions from the blog post Hidden in Plain Sight to set up the script for generating images with hidden text in `src/visualize.py`. Ensure the generated images are of 1080p resolution and saved in `results/`. Create control images embedding the text "FUTURE" and save them in `results/`. Please manually verify that the hidden text is embedded in the images.*

### Requirements

■ **R0**
**Criteria:** *Follow the instructions from the blog post Hidden in Plain Sight to set up the script for generating images with hidden text in `src/visualize.py`.*
**Dependencies** → {}

■ **R1**
**Criteria:** *Ensure the generated images are of 1080p resolution and saved in `results/`.*
**Dependencies** → {**R0**}

■ **R2**
**Criteria:** *Create control images embedding the text "FUTURE" and save them in `results/`.*
**Dependencies** → {**R1**}

### Preferences (Optional)

■ **P0**
**Criteria:** *The system should be capable of learning and adapting to unfamiliar technologies and tools as required.*

■ **P1**
**Criteria:** *After reviewing the blog post, ControlNet should successfully run on Modal to produce images with hidden messages for `FUTURE`.*

---

*Figure 7.* **A task example in DevAI**. This task is adapted from a real-world demo given at https://www.cognition.ai/blog/introducing-devin. As this example shows, task requirements in DevAI are structured as a Directed Acyclic Graph (DAG), with nodes representing individual requirements and directed edges showing dependencies. More examples are in Appendix G.

# B. Experiment Designs

This section outlines the experimental designs aimed at evaluating developer agents' performance, analyzing human evaluations, and comparing AI-based judging systems. The experiments are structured across three levels, as illustrated below.

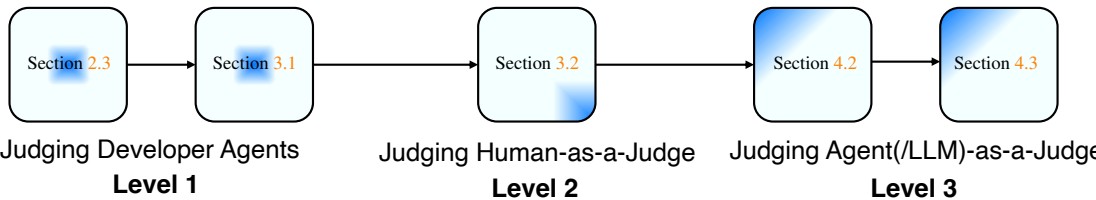

## B.1. Summary of Experiments

The experiments are categorized into three levels as follows:

**Level 1:** Human evaluation of developer agents

- *Experiment 1a:* Basic performance statistics for developer agents (Section 2.3)
- *Experiment 1b:* Human evaluations of developer agents (Section 3.1)

**Level 2:** Error and bias analysis of human evaluations

- *Experiment 2a:* Error analysis of human evaluations (Section 3.2)

**Level 3:** AI-based judging systems

- *Experiment 3a:* AI judge baselines (Section 4.2)
- *Experiment 3b:* Ablation studies for Agent-as-a-Judge (Section 4.3)

## B.2. Judges and Subjects of Evaluation

The following table summarizes the judge and the subject being evaluated in each experiment:

| Experiment | Who is the Judge? | Who is being Judged? |
| --- | --- | --- |
| Section 2.3 | *Human* | *Developer Agents* |
| Section 3.1 | *Human* | *Developer Agents* |
| Section 3.2 | *Human* | *Human* |
| Section 4.2 | (1) *LLM-as-a-Judge* | (1) *Developer Agents* |
|  | (2) *Agent-as-a-Judge* | (2) *Developer Agents* |
|  | (3) *Human* | (3) *LLM-as-a-Judge* |
|  | (4) *Human* | (4) *Agent-as-a-Judge* |
| Section 4.3 | *Human* | *Agent-as-a-Judge* |

## C. Agent-as-a-Judge Pipeline

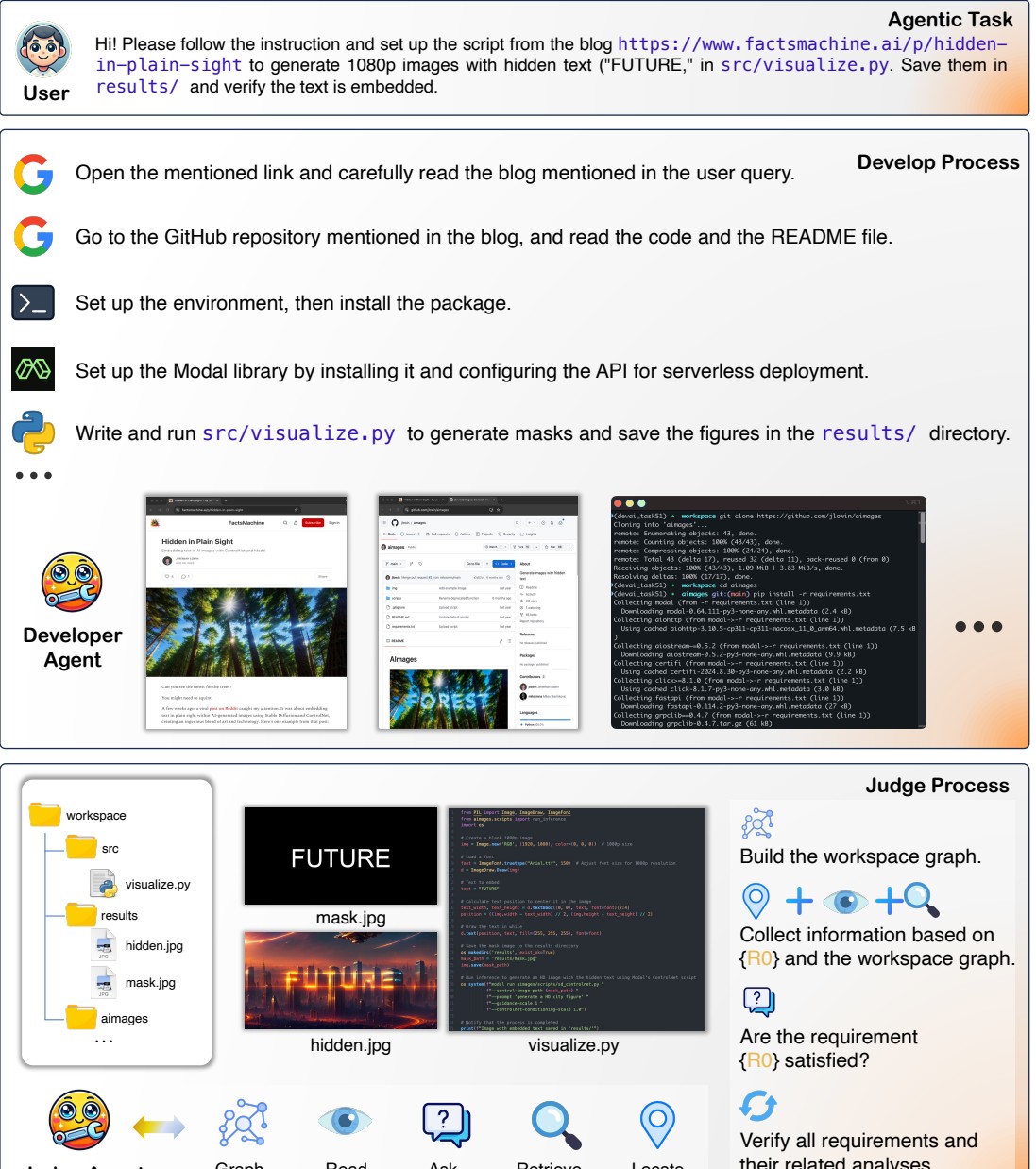

*Figure 8.* The pipelines of developer agents and judge agent.

# D. Extend Related Work

Our main paper includes mostly related works of **AI developers**, **Benchmarks for AI developments**, and **AI judges**. However, the following works contribute significantly to the community and also relate to this work. We record this work as additional related work[3].

**LLM-based Autonomous Agents** Recent developments in LLM-based agents have expanded their capabilities beyond simple task execution to more autonomous problem-solving and decision-making. AutoGPT (Gravitas, 2023) and LangChain (Chase, 2022) provide frameworks for single-agent systems that leverage external tools for more complex tasks. Similarly, research such as MetaGPT (Hong et al., 2024b), AutoGen (Wu et al., 2023) and CAMEL (Li et al., 2023) focus on role-based multi-agent communication, improving collaboration among agents. However, the challenge of maintaining coherence in agents' dialogue and preventing hallucination remains prominent (Du et al., 2024; Zhou et al., 2023). Most recently, using graphs to build agents has gained prominence. Earlier work like GPTSwarm (Zhuge et al.) and LangGraph (LangChain-AI, 2024) proposed using nodes to represent operations and edges to represent the connections between them. In GPTSwarm, multiple agents represented as subgraphs in a graph are connected by optimizable edges, and reinforcement learning is employed to optimize the edges. Following this approach, several agent frameworks have incorporated graphs into their designs (Hong et al., 2024a; Zhou et al., 2024; Qian et al., 2024b). Additionally, various optimization methods have been developed to enhance agent performance further (Wu et al., 2024; Song et al., 2024; Hu et al., 2024). In practical applications, many studies focus on understanding and interacting with GUIs (Wang et al., 2024a; Chen et al., 2024a; Yang et al., 2023; Xu et al., 2024; Tan et al.). For code generation agents (Jin et al., 2024), current research mainly emphasizes automated repair (Yang et al., 2024a; Phan et al., 2024; Tao et al., 2024), computational modular design (Khattab et al., 2024; Cheng et al.), and automated development (Tufano et al., 2024; Huang et al., 2023). Among these, open-sourced frameworks like OpenHands (Wang et al., 2024b) have gained popularity due to their strong user experience. Moreover, scientific discovery (Jansen et al., 2024; Lu et al., 2024a) and ML agents (Yang et al., 2024b) are also receiving increased attention.

**LLM-as-a-Judge** In the domain of AI evaluation and judgment, frameworks (Zheng et al., 2024; Fu et al., 2024; Chen et al.) have pioneered the use of LLMs to assess conversational agents, demonstrating how LLMs can evaluate dialogue quality and consistency. Expanding beyond dialogue, LLMs like CodeR (Chen et al., 2024b) and MASAI (Arora et al., 2024) apply similar judging principles to the code validation process, where AI systems autonomously evaluate and verify computer programs. Our work builds on these advancements by exploring how LLMs can perform more nuanced judgment tasks, further investigating their potential in decision-making across various domains. Recent research also focuses on judging LLM-as-a-Judges (Chen et al., 2024c; Bavaresco et al., 2024; Thakur et al., 2024; Dong et al., 2024; Shi et al., 2024; Raina et al., 2024).

**Coding Benchmarks** Recent advances in code generation have led to the innovation of various benchmarks to evaluate model performance (Liu et al., 2024). Early benchmarks, such as MBPP (Austin et al., 2021), HumanEval (Chen et al., 2021), and MultiPL-E (Cassano et al., 2023), focus primarily on generating simple functions. While these benchmarks are useful for evaluating the correctness of generated code, they are limited in complexity and do not fully represent the challenges encountered in real-world software development.

As the field progressed, newer benchmarks began to focus on more complex and realistic tasks. APPS (Hendrycks et al.), CodeContests (Li et al., 2022), and LiveCodeBench (Jain et al., 2024) moved toward competitive programming challenges that involve advanced algorithms and data structures. These tasks are more representative of problems encountered in coding competitions and help push models toward more sophisticated problem-solving. DS-1000 (Lai et al., 2023) was introduced to assess the skills of models with data science libraries, evaluating their ability to use APIs and execute complex data analysis workflows. Meanwhile, AgentBench (Liu et al., b) focuses on testing reasoning and decision-making abilities in interactive environments, highlighting differences in performance between commercial and open-source models.

To address real-world programming needs beyond code generation, specialized benchmarks have been created to evaluate tasks such as debugging, refactoring, and code navigation. CANITEDIT (Cassano et al., 2024), DebugBench (Tian et al., 2024), and FixEval (Haque, 2023) evaluate the ability of a model to edit and improve existing code. Additionally,

---

[3]Additionally, we were pleased to find that a recent industry blog (`https://www.cognition.ai/blog/evaluating-coding-agents`), published two weeks before our submission, shares very similar ideas and provides further evidence that the Agent-as-a-Judge could have practical applications in agent systems.

benchmarks such as SWE-Bench (Jimenez et al.) focus on resolving issues in GitHub repositories, simulating practical software development scenarios. Finally, benchmarks such as RepoBench (Liu et al., a) and RepoEval (Zhang et al., 2023) delve into the evaluation of models in large-scale, multifile codebases. These benchmarks measure the ability of language models to understand the structure of repositories and solve problems within more complex, collaborative environments.

# E. The Procedures of Creating DevAI Dataset

## E.1. Manually draft user queries

Given the execution cost of the developer agents, we collect small-scale AI tasks to ensure the practical applicability of our benchmark. Since these tasks are small-scale and well studied, which are easy to overfit in terms of task performance metrics, unlike previous benchmarks (e.g., (Huang et al., 2024)), we do not evaluate task performance as the development performance measure. Instead, we prioritize the step-by-step task-solving ability, which is essential for real-world development. Our quires are specifically designed to require the development agents to understand user intentions, solve the task in multiple steps, and adapt to unexpected step outcomes. This approach also makes our benchmark user-friendly, transparent, and better reflects real-world deployment situations. To enable effective evaluation, our queries present a specific development file structure for the developer agents to follow. To ensure that the developer agents save the files to be evaluated in the workspace, we develop constraint prompts added to the query to form an extended query. The constraint prompts guide the developer agents to save and execute the source codes, which are in line with the real-world development standard. See Appendix I for our constraint prompts.

## E.2. Set Judging Criteria

To make the evaluation of the developer agent precise, we assign to each task query a list of requirements as task milestones. The requirements are chosen so that satisfying all the requirements is a necessary condition to consider the task to be solved successively. Since our tasks are AI-centric, our queries target essential elements of AI development, including data processing, AI method, presentation of evaluation metrics, visualization, and human-computer interaction, covering the key areas that matter most in real-world scenarios. On the other hand, breaking down tasks into individual requirements also reflects the multi-step nature of code development. Importantly, to avoid ambiguity, we set the requirements to be explicit, binary, and straightforward to evaluate. To include other human predispositions, we include a list of preferences per task that covers subjective, ambiguous, or non-explicitly stated characteristics.

## E.3. Building Dependency Among Requirements

To enhance the realism of our benchmark, we analyzed the dependencies among requirements. Over the past decades, methodologies such as the KDD Process (Fayyad et al., 1996) and CRISP-DM (Wirth & Hipp, 2000) have guided ML/AI development, establishing foundational frameworks that have been further refined with the rise of AutoML (He et al., 2021).

Inspired by these methodologies, we identified a seven-step process for automated AI development tasks. This process includes critical stages such as data preprocessing, feature engineering, model selection, and hyperparameter tuning, along with essential post-development activities like metrics recording, report generation, and the development of interactive user applications. This structured approach allows us to evaluate the agent's ability to manage task dependencies and effectively navigate complex, real-world scenarios.

## E.4. Refine the dataset

Manual refinements were necessary to ensure the accuracy and clarity of DevAI. We perform two rounds of comprehensive review and edits on DevAI, each round being done by a different participant. During these manual refinements, we focus on the logical consistency of our queries and requirements, the correctness and ambiguity of the language, and the applicability of the content to the task domain. We identified a moderate number of errors in our dataset during the review.

## E.5. Analyse the dataset

We categorized each requirement based on its focus, whether it was related to the data set, the machine learning method, visualization, metrics, HCI, or data processing. While this approach provides a useful framework, it is important to recognize that these categories might overlap or miss certain nuances. Similarly, preferences were classified by how strongly they appeared in the query text, ranging from those inferred by common sense to those explicitly stated. Although this helps to organize preferences, it is worth noting that such classifications can be subjective and may not fully capture the importance of each preference in practical applications. By structuring the requirements and preferences this way, we aim to enhance the evaluation process, though flexibility and context awareness remain crucial for truly robust assessments.

## E.6. Auxiliary Information

Some of the tasks require the download of a Kaggle () data set, where a Kaggle credential is needed. Our constraint I requires an "is_kaggle_api_needed" tag to determine whether the credential is needed to be included in the extended query. We further mark each task with some tags describing the AI sub-fields related to the task, including computer vision, supervised learning, reinforcement learning, natural language processing, etc., as well as two "is_training_needed" and "is_web_navigation_needed" tags as auxiliary information. We also categorize each requirement into one of the following: (1) `dataset or environment`, (2) `data preprocessing and postprocessing`, (3) `machine learning method`, (4) `save trained model`, (4) `performance metrics`, (5) `human computer interaction`, (6) `visualization`, and (7) `other`, reflecting the nature of the requirement.

## E.7. A Json Format of Our Sample

Here, we provide a sample of the DevAI with its json format. We also provide more samples in Appendix G.

```json
{
    "name": "25_Speech_Emotion_Recognition_CNN_LSTM_RAVDESS_DL",
    "query": "I am seeking a speech emotion recognition project using a CNN-LSTM model
        with the RAVDESS dataset, which should be downloaded from Kaggle or [this Hugging
        Face link](https://huggingface.co/datasets/xbgoose/ravdess). The project should
        load the dataset and perform robust audio preprocessing (noise removal and
        normalization) and MFCC feature extraction, implemented in 'src/data_loader.py'.
        The CNN-LSTM model should be implemented in 'src/model.py'. Recognition accuracy
        should be saved in 'results/metrics/recognition_accuracy.txt', and a confusion
        matrix should be generated and saved as 'results/figures/confusion_matrix.png'.
        Additionally, a user-friendly local API should be created using Flask to allow
        users to upload audio files and receive emotion recognition results, with the
        implementation included in 'src/hci.py'.",
    "tags": [
        "Audio Processing",
        "Classification"
    ],
    "requirements": [
        {
            "requirement_id": 0,
            "prerequisites": [],
            "criteria": "The \"RAVDESS\" dataset is loaded in 'src/data_loader.py', which
                is downloaded from Kaggle or [this Hugging Face link](https://huggingface.
                co/datasets/xbgoose/ravdess).",
            "category": "Dataset or Environment",
            "satisfied": null
        },
        {
            "requirement_id": 1,
            "prerequisites": [
                0
            ],
            "criteria": "Audio preprocessing, including noise removal and normalization,
                is implemented in 'src/data_loader.py'.",
            "category": "Data preprocessing and postprocessing",
            "satisfied": null
        },
        {
            "requirement_id": 2,
            "prerequisites": [
                0,
                1
            ],
            "criteria": "MFCC feature extraction is implemented in 'src/data_loader.py'.",
            "category": "Data preprocessing and postprocessing",
            "satisfied": null
        },
```

```json
        {
            "requirement_id": 3,
            "prerequisites": [],
            "criteria": "The \"CNN-LSTM\" model is implemented in 'src/model.py'.",
            "category": "Machine Learning Method",
            "satisfied": null
        },
        {
            "requirement_id": 4,
            "prerequisites": [
                2,
                3
            ],
            "criteria": "Recognition accuracy is saved in 'results/metrics/
                recognition_accuracy.txt'.",
            "category": "Performance Metrics",
            "satisfied": null
        },
        {
            "requirement_id": 5,
            "prerequisites": [
                2,
                3,
                4
            ],
            "criteria": "The confusion matrix is generated and saved as 'results/figures/
                confusion_matrix.png'.",
            "category": "Visualization",
            "satisfied": null
        },
        {
            "requirement_id": 6,
            "prerequisites": [
                2,
                3
            ],
            "criteria": "A local API is created using \"Flask\" to allow users to upload
                audio files and receive emotion recognition results. The implementation
                should be included in 'src/hci.py'.",
            "category": "Human Computer Interaction",
            "satisfied": null
        }
    ],
    "preferences": [
        {
            "preference_id": 0,
            "criteria": "The audio preprocessing step should be robust, effectively
                reducing noise while preserving the integrity of the speech signals.",
            "satisfied": null
        },
        {
            "preference_id": 1,
            "criteria": "The local API should be user-friendly, with clear instructions
                for uploading files and interpreting results.",
            "satisfied": null
        }
    ],
    "is_kaggle_api_needed": true,
    "is_training_needed": true,
    "is_web_navigation_needed": true
}
```

## F. User experiences of code-generation agentic systems

OpenHands (Wang et al., 2024b) offers the most refined user experience, leveraging its highly interactive frontend to enable seamless user interaction and task execution. This interface allows users to engage directly with the system, resulting in a smoother and more intuitive workflow, which drives operational efficiency.

In contrast, MetaGPT (Hong et al., 2024b) excels in task decomposition through its use of Directed Acyclic Graphs (DAGs), a well-structured and scalable approach aligned with industry best practices in system modularization. This enhances its appeal for users focused on task clarity and modular breakdowns. However, in practical deployments, MetaGPT tends to be less aggressive in file management and preservation, potentially due to its core positioning as a data analysis tool, which does not prioritize persistent state management. Similarly, OpenDevin demonstrates a notable overconfidence in its code generation, frequently skipping the critical step of post-generation code execution, requiring users to intervene manually.

GPT-Pilot (Pythagora.io, 2023), praised for its detailed task delegation via over 20 specialized agents, suffers from reduced interactivity due to an overly granular division of responsibilities, resulting in a more fragmented user experience. These qualitative insights, although not fully captured by quantitative metrics, were evident through the DevAI dataset, providing key areas for improvement in user engagement and operational fluidity in future releases of these frameworks.

# G. More DevAI dataset samples

We listed more more samples in Figure 9, Figure 10, Figure 7.

# H. Human Evaluation Procedure

We recruited three AI experts from the authors to perform human evaluation on the output of agentic code generation systems. There we present the evaluation details.

**First round**  For the first round of evaluations, our three evaluators reported spending 16.5, 19.5, and 22.0 hours, respectively. To capture the bias that a human evaluator will have, the instructions given to our experts were minimal, with them only receiving a scorecard to complete for each agentic system and each task. Results that all evaluators agree on are considered trustworthy. The assumption here is that it is unlikely that all three evaluators make a mistake or have an effective bias in the same judgment. The self-reported post-hoc evaluation criteria are shown in Figure 11.

**Second round**  In the second round, the evaluators present and discuss their reasons for disagreeing with judges. In doing so, human errors are likely corrected by their peers. Discussion among evaluators also helps reduce human bias by examining each other's thought processes thoroughly. Furthermore, the consensed results are considered trustworthy given the assumption that it is unlikely that all three evaluators are convinced by the same mistake or the same cognitive bias. The three evaluators took 9.5 hours together for this second round of evaluation.

# I. Suggest Constraints

Below is a sample of constraints in JSON format that describes task-specific guidelines:

```json
{
    "generic": "This is a task that requires you to write, execute, and save source code.
        You have a hard time limit of 30 minutes to produce your programmatic solution to
        the given task. This time limit includes execution time. The quality of your
        solution will be judged based on what you left in the working folder by the time 30
         minutes expire. Additionally, the hardware you are running on is unknown, and the
        presence of a GPU is not guaranteed.",
    "is_training_needed": "Keep the time limit in mind when setting hyperparameters for
        training.",
    "is_kaggle_api_needed": "You can use the Kaggle API credentials stored in `kaggle.json`
         in your current working directory."
}
```

To address automation and security concerns, we have written code to place the `kaggle.json` file into the current workspace for each baseline during each run. However, dataset users are free to modify the solution to enable the Kaggle API for their specific developer agents.

# J. Collected Trajectories

## J.1. Schema

Below is the required JSON format for a trajectory in gray-box settings (where the trajectories can serve as input for LLM-as-a-Judge and Agent-as-a-Judge).

```
{
  "type": "array",
  "items": {
    "type": "object",
    "properties": {
      "step": {
        "type": "integer",
        "description": "The step number in the trajectory, 0-based."
      },
      "user_message": {
        "type": ["string", "null"],
        "description": "The message from the external user to the agent. If null, no
            message was sent."
      },
      "agent": {
        "type": "object",
        "properties": {
          "thought": {
            "type": "string",
            "description": "The agent's thought at this step."
          },
          "action": {
            "type": ["string", "null"],
            "description": "The agent's action sent to the environment. If null, the agent
                did not take any action, for example, when the agent has finished the
                task."
          },
          "agent_name": {
            "type": "string",
            "description": "The name of the agent that made the action."
          }
        },
        "required": ["thought", "action"],
        "description": "Everything related to the agent at this step."
      },
      "environment": {
        "type": ["string", "null"],
        "description": "The environment's (shell, python interpreter) response to the
            action submitted by the agent. If null, the environment was not involved in
            this step."
      },
      "step_usage": {
        "type": "object",
        "properties": {
          "input_tokens": {
            "type": "integer",
            "description": "The number of input tokens passed as LLM context."
          },
          "output_tokens": {
            "type": "integer",
            "description": "The number of tokens produced by the LLM."
          },
          "model": {
            "type": "string",
            "description": "The name of the LLM model used."
          },
          "cost": {
```

```
          "type": "number",
          "description": "The cost of the LLM inference, in USD."
        },
        "llm_inference_time": {
          "type": "number",
          "description": "The time taken by the LLM to generate the output tokens, in
              seconds."
        },
        "step_execution_time": {
          "type": "number",
          "description": "The time taken to make an entire step including LLM inference
              and environment execution, in seconds."
        }
      },
      "required": [
        "input_tokens",
        "output_tokens",
        "model",
        "cost",
        "llm_inference_time",
        "step_execution_time"
      ]
    },
    "accumulated_usage": {
      "type": "object",
      "properties": {
        "accumulated_cost": {
          "type": "number",
          "description": "The total cost of the trajectory up to this step, in USD."
        },
        "accumulated_time": {
          "type": "number",
          "description": "The total time taken by the agent to complete the trajectory
              up to this step, in seconds."
        }
      },
      "required": [
        "accumulated_cost",
        "accumulated_time"
      ]
    }
  },
  "required": ["step", "user_message", "agent", "environment", "step_usage", "
      accumulated_usage"]
  }
}
```

## J.2. A Sample of Trajectory

This is sample trajectories (we select 4 steps of it) in OpenHands (Wang et al., 2024b). All steps are recorded to enable replay of the trajectory.

```
{
        "step": 0,
        "user_message": "Help me develop an object detection system using the YOLOv3 model
            and the COCO dataset. Download the dataset and preprocess the images by
            resizing and normalization in 'src/data_loader.py'. Implement the YOLOv3 model
             and use Non-Maximum Suppression (NMS) to refine the results in 'src/model.py
            '. Save the detected objects to 'results/figures/', and create an interactive
            Streamlit web page in 'src/app.py' to display the detection results. Finally,
            evaluate the model's performance, including metrics such as mAP and inference
            time, and save the evaluation results to 'results/metrics/model_performance.
            txt'. The system should properly manage the launch and termination of the
            Streamlit application to prevent unnecessary resource usage.\n\nThis is a task
             that requires you to write, execute, and save source code. You have a hard
            time limit of 30 minutes to produce your programmatic solution to the given
            task. This time limit includes execution time. The quality of your solution
            will be judged based on what you left in the working folder by the time 30
            minutes expire. Additionally, the hardware you are running on is unknown, and
            the presence of a GPU is not guaranteed.\n\nKeep the time limit in mind when
            setting hyperparameters for training.\n",
        "agent": {
            "thought": "Let's break down the task into manageable steps and start
                implementing them one by one.\n\n### Steps:\n1. **Set up the directory
                structure**:\n   - Create necessary directories: 'src/' and 'results/
                figures/'.\n2. **Download and preprocess the COCO dataset**:\n   -
                Implement 'src/data_loader.py' to download, resize, and normalize images.\
                n3. **Implement the YOLOv3 model**:\n   - Implement 'src/model.py' to
                define the YOLOv3 model and apply Non-Maximum Suppression (NMS).\n4. **
                Create an interactive Streamlit web page**:\n   - Implement 'src/app.py'
                to display detection results.\n5. **Evaluate the model's performance**:\n
                  - Save evaluation metrics to 'results/metrics/model_performance.txt'.\n\
                nLet's start by setting up the directory structure and creating the
                necessary files.\n\n### Step 1: Set up the directory structure",
            "action": "Running command: mkdir -p src results/figures results/metrics\
                ntouch src/data_loader.py src/model.py src/app.py"
        },
        "environment": "\r\n[Python Interpreter: /openhands/poetry/openhands-5O4_aCHf-py3.
            11/bin/python]\nopenhands@b34322eae8da:/workspace $ touch src/data_loader.py
            src/model.py src/app.py\r\n\r\n[Python Interpreter: /openhands/poetry/
            openhands-5O4_aCHf-py3.11/bin/python]\nopenhands@b34322eae8da:/workspace $ ",
        "step_usage": {
            "input_tokens": 4331,
            "output_tokens": 220,
            "llm_inference_time": 4.445789337158203,
            "model": "gpt-4o-2024-05-13",
            "cost": 0.024955,
            "step_execution_time": 16.24921226501465
        },
        "accumulated_usage": {
            "accumulated_cost": 0.024955,
            "accumulated_time": 16.24922204017639
        }
    },
    {
        "step": 1,
        "user_message": null,
        "agent": {
            "thought": "Great! The directory structure and necessary files are created.
                Let's proceed with implementing the data loader to download and preprocess
                 the COCO dataset.\n\n### Step 2: Implement 'src/data_loader.py'\nWe will:
```

```
                \n1. Download the COCO dataset.\n2. Resize and normalize the images.\n\
                    nLet's start by writing the code for 'src/data_loader.py'.",
                "action": "Running Python code interactively: create_file('src/data_loader.py
                    ')"
        },
        "environment": "FileExistsError                          Traceback (most recent
            call last)\nCell In[1], line 1\n----> 1 create_file('src/data_loader.py')\n\
            nFile /openhands/code/openhands/runtime/plugins/agent_skills/file_ops/file_ops
            .py:257, in create_file(filename)\n    251 \"\"\"Creates and opens a new file
            with the given name.\n    252 \n    253 Args:\n    254     filename: str: The
            name of the file to create.\n    255 \"\"\"\n    256 if os.path.exists(
            filename):\n--> 257     raise FileExistsError(f\"File '{filename}' already
            exists.\")\n    259 with open(filename, 'w') as file:\n    260     file.write
            ('\\n')\n\nFileExistsError: File 'src/data_loader.py' already exists.\n[
            Jupyter current working directory: /workspace]\n[Jupyter Python interpreter: /
            openhands/poetry/openhands-5O4_aCHf-py3.11/bin/python]",
        "step_usage": {
            "input_tokens": 4675,
            "output_tokens": 84,
            "llm_inference_time": 2.136143207550049,
            "model": "gpt-4o-2024-05-13",
            "cost": 0.02463500000000004,
            "step_execution_time": 3.345384359359741
        },
        "accumulated_usage": {
            "accumulated_cost": 0.04959000000000001,
            "accumulated_time": 19.594608068466187
        }
    },
    {
        "step": 2,
        "user_message": null,
        "agent": {
            "thought": "The file 'src/data_loader.py' already exists. Let's open it and
                start writing the code to download and preprocess the COCO dataset.\n\n###
                 Step 2: Implement 'src/data_loader.py'",
            "action": "Running Python code interactively: open_file('src/data_loader.py')"
        },
        "environment": "[File: /workspace/src/data_loader.py (1 lines total)]\n(this is
            the beginning of the file)\n1|\n(this is the end of the file)\n[Jupyter
            current working directory: /workspace]\n[Jupyter Python interpreter: /
            openhands/poetry/openhands-5O4_aCHf-py3.11/bin/python]",
        "step_usage": {
            "input_tokens": 4982,
            "output_tokens": 53,
            "llm_inference_time": 2.209756851196289,
            "model": "gpt-4o-2024-05-13",
            "cost": 0.025705000000000002,
            "step_execution_time": 2.318861961364746
        },
        "accumulated_usage": {
            "accumulated_cost": 0.07529500000000001,
            "accumulated_time": 21.913471937179565
        }
    },
    ...
    {
        "step": 20,
        "user_message": null,
        "agent": {
            "thought": "It seems the indentation issue persists. Let's correct the
                indentation manually.\n\n### Corrected Code for 'src/evaluate.py'\n```
                python\nimport time\nimport numpy as np\nfrom model import yolo_v3_model,
                non_max_suppression\nfrom data_loader import preprocess_image,
                download_coco_dataset, extract_coco_dataset\nimport os\n\ndef
```

```
            evaluate_model():\n    model = yolo_v3_model()\n    download_coco_dataset
            ()\n    extract_coco_dataset()\n\n    image_dir = \"data/val2017\"\n
            image_files = [os.path.join(image_dir, f) for f in os.listdir(image_dir)
            if f.endswith('.jpg')]\n\n    total_inference_time = 0\n    num_images =
            len(image_files)\n    all_boxes = []\n    all_scores = []\n    all_classes
             = []\n\n    for image_file in image_files:\n        image =
            preprocess_image(image_file)\n        image = np.expand_dims(image, axis=0
            )\n\n        start_time = time.time()\n        predictions = model.predict
            (image)\n        end_time = time.time()\n\n        inference_time =
            end_time - start_time\n        total_inference_time += inference_time\n\n
                boxes, scores, classes = non_max_suppression(predictions[0])\n
                all_boxes.append(boxes)\n        all_scores.append(scores)\n
                all_classes.append(classes)\n\n    mean_inference_time =
            total_inference_time / num_images\n\n    # Placeholder for mAP calculation
            \n    mAP = 0.0  # This should be replaced with actual mAP calculation\n\n
                with open(\"results/metrics/model_performance.txt\", \"w\") as f:\n
                    f.write(f\"Mean Inference Time: {mean_inference_time:.4f} seconds\\
            n\")\n        f.write(f\"mAP: {mAP:.4f}\\n\")\n\nif __name__ == \"__main__
            \":\n    evaluate_model()\n'''\n\nLet's update the file with the corrected
             code.",
        "action": "Running Python code interactively: EDITED_CODE = \"\"\"import time\
            nimport numpy as np\nfrom model import yolo_v3_model, non_max_suppression\
            nfrom data_loader import preprocess_image, download_coco_dataset,
            extract_coco_dataset\nimport os\n\ndef evaluate_model():\n    model =
            yolo_v3_model()\n    download_coco_dataset()\n    extract_coco_dataset()\n
            \n    image_dir = \"data/val2017\"\n    image_files = [os.path.join(
            image_dir, f) for f in os.listdir(image_dir) if f.endswith('.jpg')]\n\n
               total_inference_time = 0\n    num_images = len(image_files)\n
            all_boxes = []\n    all_scores = []\n    all_classes = []\n\n    for
            image_file in image_files:\n        image = preprocess_image(image_file)\n
                   image = np.expand_dims(image, axis=0)\n\n        start_time = time
            .time()\n        predictions = model.predict(image)\n        end_time =
            time.time()\n\n        inference_time = end_time - start_time\n
            total_inference_time += inference_time\n\n        boxes, scores, classes =
             non_max_suppression(predictions[0])\n        all_boxes.append(boxes)\n
                    all_scores.append(scores)\n        all_classes.append(classes)\n\n
              mean_inference_time = total_inference_time / num_images\n\n    #
            Placeholder for mAP calculation\n    mAP = 0.0  # This should be replaced
            with actual mAP calculation\n\n    with open(\"results/metrics/
            model_performance.txt\", \"w\") as f:\n        f.write(f\"Mean Inference
            Time: {mean_inference_time:.4f} seconds\\\n\")\n        f.write(f\"mAP: {
            mAP:.4f}\\\\\n\")\n\nif __name__ == \"__main__\":\n    evaluate_model()\n\"
            \"\"\n\nedit_file_by_replace('src/evaluate.py', 'with open(\"results/
            metrics/model_performance.txt\", \"w\") as f:', '    with open(\"results/
            metrics/model_performance.txt\", \"w\") as f:')"
    },
    "environment": null,
    "step_usage": {
        "input_tokens": 20132,
        "output_tokens": 805,
        "llm_inference_time": 15.1565523147583,
        "model": "gpt-4o-2024-05-13",
        "cost": 0.1127350000000002,
        "step_execution_time": 15.196227073669434
    },
    "accumulated_usage": {
        "accumulated_cost": 1.2919750000000003,
        "accumulated_time": 176.99772310256958
    }
}
```

# K. Ablations of Agent-as-a-Judge

## K.1. Component Abalations

*Table 5.* **Component Ablation Studies for Agent-as-a-Judge.** We analyze the impact of sequentially adding various components (`ask`, `graph`, `read`, `locate`, `retrieve`, `planning`, `memory`, and `search`) on the performance of Agent-as-a-Judge for judging OpenHands.

| Metric | +`ask` | +`graph` | +`read` | +`locate` | +`search` | +`retrieve` | +`planning` | +`memory` |
|---|---|---|---|---|---|---|---|---|
| **Agent-as-a-Judge Performance** | | | | | | | | |
| **Alignment Rate** | 65.03% | 75.95% | 82.24% | 90.44% | 86.06% | 90.16% | 88.52% | 87.97% |

**Analysis** We designed 8 modular components for the Agent-as-a-Judge system. In the Table 5, components are added progressively from left to right. If the addition of a component led to a significant performance drop, we removed it from further iterations. Our experiments showed that adding the components `ask`, `graph`, `read`, and `locate` resulted in significant performance gains. However, when the `search` component was introduced, there was a noticeable decline in performance.

We hypothesize that the performance drop from `search` is due to its role in retrieving relevant code snippets (top-3) using BM25. The retrieval accuracy of BM25 (Robertson et al., 2009) might not have been high enough, potentially introducing noise. Moreover, as noted in Table 1, the DevAI tasks in our experiments did not generate a large amount of code. In fact, even when all code was fed into an LLM, the total content typically stayed within the maximum context length. Therefore, in simpler workspaces, `search` was less critical. However, we believe this component will become more important as the complexity of the workspace increases, making it more valuable in larger and more complex environments.

We also observed that the introduction of the `planning` mechanism did not bring a noticeable improvement in performance. This may be related to the nature of the Judge - it needs clean factual information. When `planning` is unstable, the evidence collected from different actions can become inconsistent, leading to a decline in performance. Finally, we experimented with a `memory` mechanism. Initially, we hypothesized that since DevAI tasks often involve interconnected requirements, `memory` could help track whether requirements were met. However, in practice, we saw no improvement. We suspect that the interconnected nature of the requirements may have caused biases: specifically, once a prior requirement was fulfilled, it might have overly influenced positive judgments on subsequent requirements, even if they were not fully met.

## K.2. Search Algorithms in Search Module

We initially hypothesized that the performance drop was due to the low precision of the `search` component, particularly with BM2.5. To explore this, we replaced BM2.5 with Sentence-BERT (Reimers & Gurevych, 2019) as a more advanced alternative and tested Fuzzy Search (Levenshtein, 1966) as a less precise option. However, neither improved the performance of the Agent-as-a-Judge.

hese results suggest that the performance issue is not due to BM2.5's poor search accuracy. Instead, the workspaces generated in our DevAI tasks are too simple for the `search` component to have a significant impact. In simpler workspaces, direct retrieval and evaluation are sufficient. Even though Sentence-BERT performed better than the other methods, its alignment rate (87.70%) still falls short of the configuration without the `search` component (90.44%). As workspace complexity increases, the `search` component may become more valuable.

*Table 6.* Comparisons on `Search` module with different engines.

| Search Method | Alignment Rate |
|---|---|
| BM2.5 | 86.06% |
| Sentence-BERT | 87.70% |
| Fuzzy Search | 85.52% |
| *without* Search Module | 90.44% |

## K.3. Search Algorithms in Retrieve Module

In our experiments, we found that accurately locating relevant information within a trajectory is a challenging task. Although the addition of the `retrieve` component (gray-box) did not lead to a significant improvement in performance in this specific case, its impact has been notable in other settings, such as in GPT-Pilot. As shown in Table 3, the integration of `retrieve` in GPT-Pilot brought substantial gains.

We conducted an ablation study on GPT-Pilot to optimize the retrieval of useful information at each step. Our experiments revealed that in large trajectories, truncating the final sections of the file often results in losing critical information, as the latter part of the trajectory typically contains dense information about the final development state. Truncating the beginning of the trajectory proved to be the most effective in improving the retrieval efficiency.

For individual steps, truncating the middle section worked best. This is because error messages usually appear early in the output, while the corresponding file paths and specific error locations are found towards the end. By focusing on these retrieval strategies, we can significantly enhance the performance of the `retrieve` component, particularly in complex scenarios like GPT-Pilot.

*Table 7.* Ablations on `retrieve`.

| Method | Alignment Rate |
|---|---|
| Without `retrieve` | 83.88% |
| With `retrieve` (gray-box) | 86.61% |
| Trajectory Truncate (head) | 86.61% |
| Trajectory Truncate (middle) | 85.52% |
| Trajectory Truncate (tail) | 82.51% |
| Step Truncate (head) | 86.34% |
| Step Truncate (middle) | 86.61% |
| Step Truncate (tail) | 83.88% |

# L. Prompt Demos of Agent-as-a-Judge

Here, we present some prompts used by the Agent-as-a-Judge system. Each of these prompt demos plays a crucial role in guiding the agent's behavior.

## L.1. System Prompt for Agent-as-a-Judge

```python
def get_system_prompt(language="English"):

    if language == "English":
        return """
        You are an advanced AI system serving as an impartial judge for intelligent code
            generation outputs. Your primary role is to rigorously evaluate whether the
            agent's outputs satisfy the specified requirements by thoroughly analyzing the
             provided code, data, and other relevant materials.

        You will systematically assess aspects such as datasets, model implementations,
            training procedures, and any task-specific criteria outlined in the
            requirements. Your evaluations must be objective, detailed, and based solely
            on the evidence provided.

        For each requirement, deliver one of the following judgments:

        1. <SATISFIED>: Use this if the agent's output fully meets the requirement.
            Provide a brief and precise explanation demonstrating how the specific
            criteria are fulfilled.

        2. <UNSATISFIED>: Use this if the agent's output does not meet the requirement.
            Provide a concise explanation indicating the deficiencies or omissions.

        Your assessment should reference specific elements such as code snippets, data
            samples, or output results where appropriate. Ensure that your justifications
            are clear, precise, and directly related to the criteria.

        Respond with either <SATISFIED> or <UNSATISFIED>, followed by your concise
            justification.
        """
    else:
        raise NotImplementedError(f"The language '{language}' is not supported.")
```

### L.2. System Prompt for Locate Module

```python
def get_locate_system_prompt(language="English"):
    if language == "English":
        return """
        You are an advanced AI system specializing in understanding project structures and
            determining file locations based on provided criteria.
        Your task is to locate specific files in the workspace based on the user's
            criteria and workspace information.ution problems with the files mentioned in
            the criteria.
        """

    else:
        raise NotImplementedError(f"The language '{language}' is not supported.")
```

### L.3. System Prompt for Retrieve Module

```python
def get_retrieve_system_prompt(language="English"):

    if language == "English":
        return """
        You are an advanced AI system specializing in retrieving environmental feedback
            from project execution trajectories. Your task is to analyze the provided
            trajectory data and extract information about the most relevant files
            mentioned in the given criteria.

        Focus on the following:

        1. Identify the **most recent steps** where the files directly related to the
            criteria were involved in execution, loading, or saving operations.
        2. Provide environmental feedback for these files, such as any errors, warnings,
            or issues encountered during their execution or processing.
        3. Highlight whether any problems occurred that might affect the functionality or
            success of these files in the project.

        Your output should be structured as follows:

        - **<RELEVANT STEPS>**: List the specific steps involving the relevant files,
            including any environmental feedback such as error messages, execution results
            , or other issues encountered. Each step should concisely present the key
            information needed to assess the files' execution status.

        Avoid including details about file contents or existence, as this information is
            already available. Focus solely on the environmental feedback related to the
            execution of the most relevant files.

        Your goal is to provide clear and concise information that helps determine if
            there were any execution problems with the files mentioned in the criteria.
        """
    else:
        raise NotImplementedError(f"The language '{language}' is not supported.")
```

### L.4. Prompt for Ask Module (for requirement check)

```python
def get_ask_prompt(criteria: str, evidence: str) -> str:

    return f"""
    Provided below is relevant information about the project:
    {evidence}

    Kindly perform an evaluation of the following criteria:
    {criteria}

    As per the guidelines, respond with either <SATISFIED> or <UNSATISFIED>, followed by a
        concise justification that references specific elements from the project
        information, such as code snippets, data samples, or output results.
    """
```

### L.5. Prompt for Locate Module

```python
def get_locate_prompt(criteria: str, evidence: str) -> str:

    return f"""
    Provided below is the structure of the workspace:
    {workspace_info}

    This is the criteria related to the task:
    {criteria}

    Follow the format in the example below and return only the file paths that match the
        criteria:

    Example:

    Suppose the criteria is:
    'The database functionality is implemented in `src/db.py`, and the logging system is
        defined in `src/logging.py`.'

    And the workspace information is:
    /project
    |-- src
    |    |-- db.py
    |    |-- logging.py
    |    |-- utils.py
    |-- tests
        |-- test_db.py
        |-- test_logging.py

    Based on the criteria, the following paths (no more than 5) should be returned, each
        wrapped in dollar signs ('$'):
    $/project/src/db.py$
    $/project/src/logging.py$
    """
```

# M. Judge Evidences Collected from Agent-as-a-Judge

The Agent-as-a-Judge system dynamically collects evidence throughout the development process of the code-generating agentic system to evaluate whether each requirement is satisfied. Using this auxiliary information, the judge agent decides to focus on relevant data such as code, file structures, and trajectories. This gathered evidence supports the agent's final judgment, ensuring an informed and accurate assessment of the project's compliance with its requirements. A sample of the collected evidence is shown in the following Appendix M.

# N. Analysis of failure cases

We analyzed the failure cases and identified consistent patterns across the task categories. The categories of these failure cases are summarized in Table 8. We found that AAAJ struggled most with judging cases in the *Data preprocessing and postprocessing* category, whereas it performed well in judging *Human-Computer Interaction* cases.

*Table 8.* Counts of failure cases aggregated over task categories.

| Category | Count |
|---|---|
| Data preprocessing and postprocessing | 10 |
| Dataset or Environment | 8 |
| Other | 5 |
| Machine Learning Method | 4 |
| Performance Metrics | 3 |
| Visualization | 3 |
| Human-Computer Interaction | 3 |

We collect two failure cases in Table 9 and layout their analysis below.

*Table 9.* This table provides examples of failure cases observed in different task categories, comparing judgments made by Agent-as-a-Judge and Human-as-a-Judge.

| Ex. # | Task | Req. ID | Category | Agent-as-a-Judge | Human-as-a-Judge | Criteria |
|---|---|---|---|---|---|---|
| 1 | `40_Text_Summarization_BART_CNNDailyMail_DL` | 0 | Dataset or Environment | False | True | The "CNN/Daily Mail" news dataset is used, including loading and preparing the dataset in `src/data_loader.py`. |
| 2 | `46_Speech_Recognition_DeepSpeech_LibriSpeech_DL` | 2 | Machine Learning Method | True | False | Hyperparameters such as learning rate and batch size are tuned in `src/train.py`. |

**Analysis**

- **Failure Case Example 1:** In this case, the dataset used was a synthesized one generated by the OpenHands CodeAct agent. Human evaluators could quickly identify this discrepancy, but the agent-as-a-judge, having only checked the file path and content, was misled into believing it was the genuine CNN/DailyMail dataset.

- **Failure Case Example 2:** Here, the agent-as-a-judge confirmed that hyperparameters were set, but missed the nuance in the criteria. The requirement implied that the learning rate and batch size should dynamically adjust in `src/train.py`, something human evaluators were able to detect.

## O. Sensitivity w.r.t the choice of the backend LLM

We have run an ablation experiment to determine how different LLM backends affect the performance of Agent-as-a-Judge. The results are summarized in Table 10.

*Table 10.* This table reports alignment percentages between Agent-as-a-Judge and Human-as-a-Judge for different backend LLMs.

| Model | Version | # Param. | Alignment (%) |
|---|---|---|---|
| LLAMA, (Touvron et al., 2023) | 3.2 | 90B | 87.76% |
| Qwen, (Bai et al., 2023) | Coder 2.5 | 32B | 88.73% |
| ChatGPT, (OpenAI, 2023) (This work) | gpt-4o-2024-0513 | Unknown | 90.16% |
| Claude, (Anthropic, 2024) | claude-3-5-sonnet-20241022 | Unknown | 92.95% |

These results allow us to conclude that the backbone does have a noticeable effect on the alignment but a relatively marginal one. We found that Claude's results are better than GPT-4o's that we used throughout our experiments. We hypothesize, this is because `claude-3-5-sonnet-20241022` has been trained with strong function calling skills and agentic features.

## P. Additional Human Evaluation Details

To check the reliability of our consensus results, we conducted an additional study with ten evaluators. All ten of our additional participants are current M.Sc. and Ph.D. students in AI-related fields with no direct relation to this work. The 7 random samples they evaluated were selected from the answers produced by OpenHands. The additional participants reported an average completion time of 1.13 hours each to evaluate all 7 samples. The additional ten participants self-reported an average completion time of 1.13 hours. This means that they took an average of 9.67 minutes to evaluate one task, which is similar to our three main evaluators, who self-reported taking an average of 10.48 minutes per task. Each of the ten additional participants was compensated for their time, with an average compensation of 15.20 USD.

In this additional evaluation, the alignment of the majority vote (with 10 experts) with our previous majority vote results (with 3 experts) was 97.67%. This consistency reinforces the reliability of our initial evaluation and suggests that AAAJ's performance is comparable to that of a broader human panel. We observe that compared to the previous results of the majority vote with three human experts (92.85% for the same 7 tasks, totaling 42 requirements), the extended study achieved a modest improvement of 2.38% in alignment, which demonstrates that more experts involved in the majority vote can improve the alignment rate. After checking the disagreement between the majority vote results from the larger panel of experts and our previous consensus results, we found that the consensus results are accurate. We suggest that this is due to the natural features of human brainstorming may be more effective in correcting errors and biases.

## Task 13: Style Transfer with Perceptual Loss in PyTorch

### Query

*Please create a PyTorch Perceptual Loss project for image style transfer (refer to this paper: Perceptual Losses for Real-Time Style Transfer). You can build the Perceptual Loss Network using VGG16 in* `src/model.py`*. The project should combine content and style images, allow smooth adjustment of style intensity by tuning the weights of style loss and content loss, and save the stylized images in* `results/figures/`*. Additionally, log the processing time to* `results/processing_time.txt`*, and save the intermediate results of the style transfer process to* `results/figures/intermediate_results.png`*. For testing, input a famous content image (Mona Lisa) from this link and a famous style image (The Starry Night) from this link, and generate a style-transferred image. Save the content, style, and style-transferred images to* `data/content.jpg`*,* `data/style.jpg`*, and* `results/figures/`*, respectively. The project should efficiently handle high-resolution images without excessive processing time.*

### Requirements

- **R0**
  **Criteria:** *A famous content image is inputted for testing, downloaded from this link and saved to* `data/content.jpg`*.* **Dependencies** → {}

- **R1**
  **Criteria:** *A famous style image is inputted for testing, downloaded from this link and saved in* `data/style.jpg`*.* **Dependencies** → {}

- **R2**
  **Criteria:** *The Perceptual Loss model is implemented in PyTorch and loaded in* `src/model.py`*.* **Dependencies** → {}

- **R3**
  **Criteria:** *Stylized images are saved to the specified folder* `results/figures/`*.* **Dependencies** → {**R0, R1, R2**}

- **R4**
  **Criteria:** *Style intensity is adjusted by tuning the weights of style loss and content loss in* `src/model.py`*.* **Dependencies** → {**R0, R1, R2**}

- **R5**
  **Criteria:** *Processing time is recorded and saved as* `results/processing_time.txt`*.* **Dependencies** → {**R0, R1, R2, R3, R4**}

- **R6**
  **Criteria:** *Intermediate results of style transfer are saved as* `results/figures/intermediate_results.png`*.* **Dependencies** → {**R0, R1, R2, R3, R4**}

### Preferences (Optional)

- **P0**
  **Criteria:** *The style transfer process should allow for smooth adjustment of style intensity, making the stylized image visually appealing.*

- **P1**
  **Criteria:** *The project should handle high-resolution images efficiently without excessive processing time.*

*Figure 9.* **An Example Task in DevAI**: Task 13.

## Task 19: Time Series Forecasting with Seq2Seq LSTM on Rossmann Store Sales

### Query

*Develop a sales forecasting system using a sequence-to-sequence model based on LSTM with the Rossmann Store Sales dataset, downloading it from Kaggle here and loading it in* `src/data_loader.py`. *Split the data into training and testing sets and save them in* `src/data_loader.py`. *Apply a sequence-to-sequence model based on* `LSTM` *and save the trained model under the* `models/saved_models/` *directory. Save the forecast results as* `results/figures/forecast_results.png`. *Save a comparison plot between the predicted and actual values to* `results/figures/comparison_plot.png`. *Generate an HTML report that includes the prediction results and comparison plots, with some interactive elements for exploring different forecast horizons, and save it as* `results/report.html`. *Ensure the model is tuned to capture seasonal trends in the sales data.*

### Requirements

■ **R0**
**Criteria:** *The Rossmann Store Sales dataset is used, potentially downloaded from (this link) and loaded in* `src/data_loader.py`. **Dependencies** → {}

■ **R1**
**Criteria:** *The data is split into training and testing sets and implemented in* `src/data_loader.py`. **Dependencies** → {**R0**}

■ **R2**
**Criteria:** *A sequence-to-sequence model based on LSTM is used. The trained model should be saved under* `models/saved_models/`. **Dependencies** → {**R1**}

■ **R3**
**Criteria:** *The forecast results are plotted and saved as* `results/figures/forecast_results.png`. **Dependencies** → {**R1, R2**}

■ **R4**
**Criteria:** *A comparison plot of predicted vs. actual values is saved as* `results/figures/comparison_plot.png`. **Dependencies** → {**R1, R2, R3**}

■ **R5**
**Criteria:** *An HTML report containing forecast results and comparison plots is generated and saved as* `results/report.html`. **Dependencies** → {**R1, R2, R3, R4**}

■ **R6**
**Criteria:** *The HTML report should include interactive elements that allow users to explore different forecast horizons.* **Dependencies** → {**R5**}

### Preferences (Optional)

■ **P0**
**Criteria:** *The model should be tuned to capture seasonal trends in the sales data for more accurate forecasting.*

*Figure 10.* **An Example Task in DevAI**: Task 19.

---

**Self-reported Post-hoc Evaluation Criteria after round one**

### Evaluator $231a$

EXECUTED SUCCESSFULLY: Yes [ ] / No [ ]
1. Must be checked based on the overall completeness of the task, based on looking at the code, the artifacts, and the trajectory.
2. The training has finished, the model snapshot and the metrics breakdown have been saved, and at least one artifact of required analytics has been produced - mark as successful. If some analytic artifacts are missing but not all, mark as successful.
3. No need to run the code.
4. If training was finished but on fake data, mark as successful.
Requirements:
Marking a requirement as satisfied must be made for this specific requirement disregarding the dependency list. If a file (code, image, snapshot) is there but is empty or without any meaningful content - mark as No.
1. Code:
The functionality must be in a file with the requested path.
The real data is replaced by simple synthetic - not satisfied.
2. Visualization/Reports:
The contents must be there and make sense even if not perfect from the ML/DS point of view.
3. Snapshots:
If a binary snapshot is not empty, mark as Yes.

### Evaluator $38bb$

EXECUTED SUCCESSFULLY: Yes [ ] / No [ ]
An output is marked yes if none of the following is satisfied.
1. The time spent is close to the time limit.
2. The last environmental message includes an error.
3. The last thought indicates that the task is completed.
4. The last step of the trajectory is incomplete.
Requirements:
If a required is unsatisfied only because of an unsatisification of a previously marked unsatisfied requirment, then judge it based on the assumption that a minimum implementation satisfies the previous requirment exists.
1. Code:
Mark yes if the code executes and does the required function. If no entrypoint is given, the evaluator will set an entrypoint. If the code is not executable due to previous unsatisfied requirements, then it is judged based on eye-checking.
2. Visualization/Reports:
Mark yes if the visualization or report exists in a right path and the content aligns the requirement.
3. Snapshots:
Mark yes if the snapshot exists in a right path and is not empty.

### Evaluator $cn9o$

I evaluated everything based on whether the requirement was satisfied verbatim, using my own professional judgement when there was ambiguity. If there was a data folder and a look at the trajectory indicated it was real, I allowed it to be loaded directly. If there were results and no clear source code making them, I considered the task incomplete. I didn't consider a dummy data path to be correct (GPT-Pilot used a lot of these). To determine if something was executed correctly, I skimmed the end of the logs (e.g., trajectory) for any obvious signs of an error. If there wasn't any, I said it was correctly executed. I ignored prerequisites while evaluating and did not execute anything, instead just eyeballing the code for correctness. I was lenient in what I considered to be sufficient in terms of the more vague requirements (e.g., if the preprocessing had to include scaling and rotation, just those two would be sufficient to consider it done).

*Figure 11.* Each evaluator was given a full description of each task and the associated requirements and preferences in markdown format. They were then asked, for each workspace and trajectory generated by each of the agents on each of the task, whether (1) the agent successfully finished its execution cycle and (2) which of the requirements were satisfied. After the evaluation was complete, the evaluators were asked to self-report the nuances of their evaluation.

```
 1   >>> [Reference] The Following Is the Orignal User Query:
 2
 3   Please help me build a system for sentiment analysis on tweets using the Sentiment140 dataset available from Hugging Face. Load the Sentiment140 dataset and,
     when loading the data, clean it by removing stop words, punctuation, and special characters, all in `src/data_loader.py`. Use Word2Vec or GloVe for text
     vectorization. This should occur in `src/data_loader.py`. Next, implement and train an SVM classifier in `src/model.py`. Finally, write a report of the accuracy
     of the classifier to `results/metrics/accuracy_score.txt`. Ideally, the report should be easily interpretable.
 4
 5   >>> [Key Evidence] The Following Is the Project and Workspace Structure Information:
 6
 7   ┌─ Project Tree ───────────────────────────────────────────────────────────────────────────────────────────────────────────────────────────────┐
 8   │                                                                                                                                                │
 9   │  Workspace Path: OpenHands/workspaces/06_Sentiment_Analysis_SVM_Sentiment140_ML                                                                 │
10   │  Total Nodes: 5                                                                                                                                 │
11   │                                                                                                                                                │
12   │  Project Structure                                                                                                                             │
13   │  ├── .                                                                                                                                          │
14   │  ├── results                                                                                                                                    │
15   │  ├── results/metrics                                                                                                                            │
16   │  ├── results/model                                                                                                                              │
17   │  └── src                                                                                                                                        │
18   │       ├── data_loader.py                                                                                                                        │
19   │       └── model.py                                                                                                                              │
20   │                                                                                                                                                │
21   └────────────────────────────────────────────────────────────────────────────────────────────────────────────────────────────────────────────┘
22
23
24   >>> [Reference] The Following Is Relevant Search Evidence:
25
26   ┌─ Code Snippet ───────────────────────────────────────────────────────────────────────────────────────────────────────────────────────────────┐
27   │                                                                                                                                                │
28   │  File: OpenHands/workspaces/06_Sentiment_Analysis_SVM_Sentiment140_ML/src/data_loader.py                                                        │
29   │  Lines: [28, 28]                                                                                                                                │
30   │  Identifier: ref                                                                                                                                │
31   │  Category: function                                                                                                                             │
32   │                                                                                                                                                │
33   │    28      dataset = load_and_preprocess_data()                                                                                                 │
34   │    29                                                                                                                                           │
35   │                                                                                                                                                │
36   └────────────────────────────────────────────────────────────────────────────────────────────────────────────────────────────────────────────┘
37   ┌─ Code Snippet ───────────────────────────────────────────────────────────────────────────────────────────────────────────────────────────────┐
38   │                                                                                                                                                │
39   │  File: OpenHands/workspaces/06_Sentiment_Analysis_SVM_Sentiment140_ML/src/model.py                                                              │
40   │  Lines: [10, 10]                                                                                                                                │
41   │  Identifier: ref                                                                                                                                │
42   │  Category: function                                                                                                                             │
43   │                                                                                                                                                │
44   │    10      dataset = load_and_preprocess_data()                                                                                                 │
45   │    11                                                                                                                                           │
46   │                                                                                                                                                │
47   └────────────────────────────────────────────────────────────────────────────────────────────────────────────────────────────────────────────┘
48   ┌─ Code Snippet ───────────────────────────────────────────────────────────────────────────────────────────────────────────────────────────────┐
49   │                                                                                                                                                │
50   │  File: OpenHands/workspaces/06_Sentiment_Analysis_SVM_Sentiment140_ML/src/data_loader.py                                                        │
51   │  Lines: [21, 26]                                                                                                                                │
52   │  Identifier: def                                                                                                                                │
53   │  Category: function                                                                                                                             │
54   │                                                                                                                                                │
55   │    21 def load_and_preprocess_data():                                                                                                           │
56   │    22     # Load the Sentiment140 dataset                                                                                                       │
57   │    23     dataset = load_dataset('sentiment140', split='train[:1%]', trust_remote_code=True)                                                    │
58   │    24     # Clean the text data                                                                                                                 │
59   │    25     dataset = dataset.map(lambda x: {'text': clean_text(x['text'])})                                                                      │
60   │    26     return dataset                                                                                                                        │
61   │                                                                                                                                                │
62   └────────────────────────────────────────────────────────────────────────────────────────────────────────────────────────────────────────────┘
63
64
65   >>> [Key Evidence] The Following Is the Contents of the Mentioned Files:
66
67   Content of OpenHands/workspaces/06_Sentiment_Analysis_SVM_Sentiment140_ML/src/data_loader.py:
68   ```
69   import re
70   import nltk
71   from datasets import load_dataset
72   from nltk.corpus import stopwords
73   from nltk.tokenize import word_tokenize
74
75
76   nltk.download('stopwords')
77   nltk.download('punkt')
78
79   def clean_text(text):
80       # Remove special characters and punctuation
81       text = re.sub(r'[^a-zA-Z\s]', '', text)
82       # Tokenize the text
83       tokens = word_tokenize(text)
84       # Remove stop words
85       tokens = [word for word in tokens if word.lower() not in stopwords.words('english')]
86       return ' '.join(tokens)
87
88   def load_and_preprocess_data():
89       # Load the Sentiment140 dataset
90       dataset = load_dataset('sentiment140', split='train[:1%]', trust_remote_code=True)
91       # Clean the text data
92       dataset = dataset.map(lambda x: {'text': clean_text(x['text'])})
93       return dataset
94
95   if __name__ == "__main__":
96       dataset = load_and_preprocess_data()
97   print(dataset[0])
98   ```
99
100  >>> [Reference] The Following Is Development Trajectory Evidence:
101
102  ┌─ Relevant Steps in Trajectory ───────────────────────────────────────────────────────────────────────────────────────────────────────────────┐
103  │                                                                                                                                                │
104  │  The following environment feedback is provided for reference only and does                                                                    │
105  │  not serve as decisive evidence. For final judgment, please rely primarily                                                                     │
106  │  on local file contents.                                                                                                                       │
107  │                                                                                                                                                │
108  │  – **Step 21**: The `src/data_loader.py` script was executed to preprocess                                                                      │
109  │  the Sentiment140 dataset. The environment feedback indicated successful                                                                        │
110  │  execution of the script, with the dataset being loaded and preprocessed.                                                                       │
111  │  The script included downloading necessary NLTK data packages, such as                                                                          │
112  │  stopwords and punkt, which are essential for text preprocessing tasks like                                                                     │
113  │  removing stop words and tokenization.                                                                                                          │
114  │                                                                                                                                                │
115  │  – **Step 23**: The `src/data_loader.py` script was run again to verify the                                                                     │
116  │  preprocessing steps. The environment feedback confirmed that the                                                                               │
117  │  preprocessing was completed successfully, and the first example from the                                                                       │
118  │  dataset was printed, indicating that the dataset was loaded and processed                                                                       │
119  │  correctly.                                                                                                                                     │
120  │                                                                                                                                                │
121  │  Overall, the preprocessing in `src/data_loader.py` was executed without                                                                        │
122  │  errors, and the necessary steps for cleaning the dataset, such as removing                                                                      │
123  │  stop words, punctuation, and special characters, were implemented and                                                                          │
124  │  verified.                                                                                                                                      │
125  │                                                                                                                                                │
126  └────────────────────────────────────────────────────────────────────────────────────────────────────────────────────────────────────────────┘
127
128
129  >>> [Reference] The Following Is Historical Judgments:
130
131  – Requirement 1: The "Sentiment140" dataset, available from "Hugging Face," is obtained in `src/data_loader.py`.
132    Satisfied: True
```

