# OpenReview forum: "Agent-as-a-Judge: Evaluate Agents with Agents"
_ICML.cc/2025/Conference — ICML 2025 poster_

### Official Review · Reviewer_Vz85 · 2025-03-09

**Overall Recommendation:** 4

**Summary:**

The paper introduces a novel framework where agentic systems are evaluated by other agentic systems. In contrast to evaluating agentic systems with human efforts, the proposed method leverages agents to provide feedback throughout the task solving process. The paper first propose DevAI benchmark to generate task solving trajectories with different agentic systems. Experiment results suggest that agent-as-a-judge closely aligns with human judgement, while dramatically reducing evaluation time and cost.

**Claims And Evidence:**

Yes.

**Essential References Not Discussed:**

N/A

**Experimental Designs Or Analyses:**

Yes. The experiment design is technically sound.

**Methods And Evaluation Criteria:**

Yes.

**Other Comments Or Suggestions:**

N/A

**Other Strengths And Weaknesses:**

Strengths:
1. The idea is novel and technically sound. To the best of my knowledge, this is the first paper that proposes to leverage agentic systems to evaluate agents.
2. The experimental results are convincing, demonstrating that the Agent-as-a-Judge paradigm aligns closely with human evaluators while dramatically reducing evaluation time and cost.
3. The authors provide a detailed account of the design process, including trial-and-error experiences that guided the development of the agentic evaluation system. The ablation study also offers clear insights into the contribution of each component.
4. The writing is well-written and easy to follow.

Weaknesses:
1. While the Agent-as-a-Judge paradigm is novel, the specific agentic system designed for evaluation appears to be closely tailored to the DevAI benchmark. This raises questions about the generalizability of the approach to other evaluation benchmarks.
2. The research primarily focuses on evaluating coding agentic systems. Although this narrow focus is understandable given the scope of the work, it leaves open the question of how well the proposed method would perform in other areas, potentially limiting its broader impact.

**Questions For Authors:**

1. In the study, Agent-as-a-Judge incurs only a slightly lower cost than LLM-as-a-Judge, even though the agentic system involves calling different module and accumulating context—which would typically lead to more API calls and cost. Could you explain why the additional processing does not result in substantially higher costs?
2. Could you clarify the definitions of “independent tasks (I)” and “tasks considering task dependencies (D)”? Specifically, in the (D) setting, does the failure of a prerequisite task automatically lead to the failure of the dependent task, or is there an alternative evaluation pipeline?

**Relation To Broader Scientific Literature:**

1. The paradigm proposed in this paper offers a unique pespective in evaluating agentic systems with higher efficiency, accuracy and lower cost.
2. The DevAI benchmark reflects the realistic AI development requirements from AI practitioners. It can serve as a useful testbed for evaluating agentic systems designed for coding. Moreoever, it mirrors broader trends in the field where there is a push toward understanding the full decision-making process of agentic systems rather than relying on binary outcome metrics.

**Theoretical Claims:**

N/A. No theoretical claims involved in this paper.

---

> ### Author Rebuttal · Authors · 2025-03-31
>
> **We sincerely thank you for the valuable time and encouraging words. Below, we address your concerns and provide additional clarifications to strengthen our paper.**
>
> ---
>
> > W1&W2. While the Agent-as-a-Judge paradigm is novel, the specific agentic system designed for evaluation appears to be closely tailored to the DevAI benchmark. This raises questions about the generalizability of the approach to other evaluation benchmarks; The research primarily focuses on evaluating coding agentic systems. Although this narrow focus is understandable given the scope of the work, it leaves open the question of how well the proposed method would perform in other areas, potentially limiting its broader impact.
>
> Thank you for this insightful point. While, the **concept** of "Agent-as-a-Judge" is inherently general, we acknowledge that, in this work, we focused on code tasks and we do not claim that our specific **implementation** is universally general.
>
> However, this does not mean that achieving such generality will necessitate overcoming significant bottlenecks. Our implementation is modular, comprising components such as `graph`, `read`, `locate`, `ask`, and `retrieve`. By modifying the prompts and workflow, it should not be difficult to transfer the system to other domains---a common practice in agentic projects. For example, the `read` module already supports multimodal data (code, text, images, etc.), and the `ask` module can be adapted to verify arbitrary requirement criteria.
>
> As you'll be aware, the field of agentic systems is moving quite quickly. Indeed, several other papers have already directly
>  implemented agent-as-a-judge for other applications, like mobile apps [1], long documents [2], bioinformatics [3], github issue [4]:
>
> [1] AutoEval: A Practical Framework for Autonomous Evaluation of Mobile Agents
>
> [2] Agent-as-Judge for Factual Summarization of Long Narratives
>
> [3] BioAgents: Democratizing Bioinformatics Analysis with Multi-Agent Systems
>
> [4] Unveiling Pitfalls: Understanding Why AI-driven Code Agents Fail at GitHub Issue Resolution
>
> However, we strongly agree with the reviewers that a more general agent-as-a-judge implementation is urgently needed—for evaluating current popular agent frameworks (such as the general agent Manus [1] or giving feedback (or rewards) to training foundation models with agentic features). As this is a cumbersome task, we believe this would merit its own follow up paper.
>
> [1] https://manus.im/
>
>
> > Q1. In the study, Agent-as-a-Judge incurs only a slightly lower cost than LLM-as-a-Judge, even though the agentic system involves calling different modules and accumulating context—which would typically lead to more API calls and cost. Could you explain why the additional processing does not result in substantially higher costs?**
>
> In principle,  Agent-as-a-Judge will cost more. But we have implemented several strategies to reduce these expenses. For example, we limit input token length by retrieving only the most relevant parts and truncating excessively long trajectories. Additionally, we use free embedding models such as BM2.5 and Sentence-BERT to avoid further cost. These are the kinds of optimizations normal in agentic system design but are not part of LLM-as-a-Judge.
>
>
> > Q2. Could you clarify the definitions of “independent tasks (I)” and “tasks considering task dependencies (D)”? Specifically, in the (D) setting, does the failure of a prerequisite task automatically lead to the failure of the dependent task, or is there an alternative evaluation pipeline?
>
> Certainly. We define:
>
> - **Independent Tasks (I):** Each task is evaluated in isolation, assuming no interdependencies. A task's success or failure has no bearing on other tasks. This treats DevAI as a set of 365 independent tasks.
> - **Tasks Considering Dependencies (D):** Here, requirements are structured as a directed acyclic graph (DAG). If a prerequisite requirement is unsatisfied, all downstream dependent requirements are marked as failed. This models realistic task pipelines where a failure early in the chain precludes successful completion of dependent components. This treats DevAI as a set of 55 multi-step tasks.
>
> This propagation rule is enforced systematically in our evaluation to reflect authentic multi-stage task structures. We will make this logic more explicit in the revision.
>
> ---
>
> **We sincerely appreciate the reviewers' thoughtful engagement and constructive criticism, which substantially enhances the clarity and quality of our paper.**

---

> > ### Comment · Reviewer_Vz85 · 2025-04-03
> >
> > I thank the authors for the thoughtful response. My questions are addressed and I have revised my score.

---

> > > ### Author Response · Authors · 2025-04-05
> > >
> > > Thanks to the reviewer for improving the scores and your professional suggestions during the review and rebuttal period.

---

### Official Review · Reviewer_qsEL · 2025-03-09

**Overall Recommendation:** 5

**Summary:**

This paper proposes the Agent-as-a-Judge framework, which leverages agentic systems to evaluate other AI agents, addressing the limitations of current evaluation methods that either ignore intermediate steps or are too labor-intensive. To showcase this approach, the authors introduce DevAI, a new benchmark consisting of 55 realistic AI code-generation tasks with detailed, hierarchical requirements designed explicitly for comprehensive agentic evaluation.The author then benchmark three leading open-source agentic systems: MetaGPT, GPT-Pilot, and OpenHands. The author finds that the proposed Agent-as-a-Judge framework closely aligns with human consensus evaluations, significantly outperforming traditional LLM-based evaluations, and dramatically reduces evaluation time and cost.

**Claims And Evidence:**

To me, most of the claims made in this work are supported by clear and convincing evidences. Two most important claims made in this paper are:
(1) Current evaluation methods for agentic AI systems either focus on final outcomes or require extensive human effort: This make senses to me and the author also details the high human labor costs encountered in manual evaluations.
(2) Agent-as-a-Judge framework significantly improves evaluation accuracy: This have benn demonstrated by extensive experiments.

**Essential References Not Discussed:**

N/A

**Experimental Designs Or Analyses:**

I checked all the experiments. most of them looks good to me.

**Methods And Evaluation Criteria:**

Yes. The experiments design make senses to me. The criterias used in this paper includes Alignment Rate, Judge Shift, Cost and Time Efficiency. These criterias are appropriate, well-chosen, and convincingly argued as essential dimensions for comprehensive agent evaluation.

**Other Comments Or Suggestions:**

Some hyperparameters might plays critical role in determining how well a model performs, It would be great to put them in Appendix.

**Other Strengths And Weaknesses:**

**Strengths**

1. The paper are well-written and easy to follow. To me, I could easy follow the logic of this work and the whole paper is written in a high standard.

2. I think the paper pointed out and addresses an important problem in this area, i.e., how to efficiently evaluate an agentic systems without involving too much human efforts. I believe using agentic systems to perform the evaluations are a good practice.

3. Rigorous Experimental Validation. Some new metrics are proposed in the experimental sections. They all make senses and looks good to me. To me, these metrics are appropriate tools to perform the evaluations in this specific areas.

**Weaknesses**

1. The experiments are only conducted on agent coding domain. Could the author specific why you choose this domain to perform the experiments? Since the agentic AI could also be applied on other domains not only limited to coding. If the author could elaborate it that would be great.

2. The evaluation approach relies heavily on advanced underlying language models (e.g., GPT-4o). I want to know whether the method highly depends on advanced model?

**Questions For Authors:**

1. Does the method highly depend on advanced model?

2.  Why you choose coding as the domain to perform evaluations?

**Relation To Broader Scientific Literature:**

N/A

**Theoretical Claims:**

N/A

---

> ### Author Rebuttal · Authors · 2025-03-31
>
> **We greatly value your time, insights, and positive feedback which encourage us a lot. Below, we will carefully address your specific questions and suggestions individually.**
>
> ---
>
> > W1&Q2. The experiments are only conducted on agent coding domain. Could the author specific why you choose this domain to perform the experiments? Since the agentic AI could also be applied on other domains not only limited to coding. If the author could elaborate it that would be great. Why was coding chosen as the domain for evaluations?
>
>
> We focused on code generation as it has clear intermediate objectives and is a promising application of agentic systems, but the concept of Agent-as-a-Judge is domain-agnostic. Even the modular implementation of our version of Agent-as-a-Judge (e.g. graph parsing, search, read, etc.) already support multiple data modalities, including code, text, images, and videos. We expect that extending Agent-as-a-Judge to other domains might just require one to potentially recombind our modules (potentially adding more functions their own).
>
> As you'll be aware, the field of agentic systems is moving quite quickly. Indeed, several other papers have already directly
>  implemented agent-as-a-judge for other applications, like mobile apps [1], long documents [2], bioinformatics [3], github issue [4]:
>
> [1] AutoEval: A Practical Framework for Autonomous Evaluation of Mobile Agents
>
> [2] Agent-as-Judge for Factual Summarization of Long Narratives
>
> [3] BioAgents: Democratizing Bioinformatics Analysis with Multi-Agent Systems
>
> [4] Unveiling Pitfalls: Understanding Why AI-driven Code Agents Fail at GitHub Issue Resolution
>
>
>
> > W2&Q1.  The evaluation approach relies heavily on advanced underlying language models (e.g., GPT-4o). I want to know whether the method highly depends on advanced model?  Does the method highly depend on advanced model?
>
> Thank you for raising this important question. Here we refer to our relevant experimental analysis:
>
>
>
> | **Model**                        | **Version**                | **\# Params** | **Alignment (%)** |
> |----------------------------------|----------------------------|---------------|-------------------|
> | **LLaMA** (Meta, 2024) | 3.2                        | 90B           | 87.76%            |
> | **Qwen**  (Alibaba, 2024) | Coder 2.3                  | 34B           | 88.73%            |
> | **ChatGPT** (OpenAI, 2024)       | gpt-4o-2024-0513     | Unknown       | 90.16%            |
> | **Claude** (Anthropic, 2024)     | claude-3.5-sonnet-20241022 | Unknown       | 92.95%            |
>
> As shown, claude-3.5-sonnet-20241022 and gpt-4o-2024-0513  achieve alignment above 90%, while LLaMA and Qwen remain competitive with alignment above 85%. This is promising, however it---like everything in modern agentic systems---does indicate that the performance is dependent on the strength of the underlying mobel. This is also positive as it means we can expect the alignment to increase as foundation models advance. For example, Claude 3.5 is widely recognized as a milestone LLM for agentic systems (it helps a lot for agentic companys like Cursor [1], Bolt [2], etc), and it plays a crucial role in enhancing the performance of Agent-as-a-Judge. Together with recent advances like DeepSeek, these performance improvements may come at a negligible cost.
>
> [1] https://www.cursor.com/
>
> [2] https://bolt.new/
>
> ---
>
> **Thank you again for your careful review and valuable suggestions.**

---

### Official Review · Reviewer_PH3V · 2025-03-11

**Overall Recommendation:** 3

**Summary:**

This paper introduces the Agent-as-a-Judge framework for evaluating agentic systems using other agentic systems, extending the LLM-as-a-Judge paradigm by incorporating feedback on intermediate task-solving steps. The authors present DevAI, a benchmark of 55 realistic AI code generation tasks with 365 hierarchical requirements, and evaluate three leading code-generating systems (MetaGPT, GPT-Pilot, and OpenHands). Their results demonstrate that Agent-as-a-Judge significantly outperforms LLM-as-a-Judge , matches or exceeds the reliability of human evaluators, and dramatically reduces evaluation costs and time. The authors conduct extensive experiments showing how different components contribute to the framework's effectiveness, presenting a promising approach for scaling evaluation of increasingly complex agentic systems.

**Claims And Evidence:**

The paper's primary claims about Agent-as-a-Judge outperforming LLM-as-a-Judge and approaching human-level evaluation quality are well-supported by the empirical evidence presented in Table 3, showing strong alignment rates with human consensus evaluations. The cost and time efficiency claims are also quantitatively supported with specific percentage reductions.

**Essential References Not Discussed:**

No.

**Experimental Designs Or Analyses:**

The ablation studies effectively isolate component contributions, while the alignment metrics provide a reasonable basis for comparison. However, some methodological concerns exist: (1) the additional ten evaluators only assessed 7 of the 55 tasks, limiting validation scope; (2) all evaluations focus exclusively on code generation tasks, restricting claims about broader applicability; and (3) the black-box vs. gray-box comparisons introduce inconsistencies in how systems access trajectory information.

**Methods And Evaluation Criteria:**

The evaluation would be stronger with a larger task corpus beyond 55 examples and demonstrations in domains outside code generation, as the current approach primarily validates the method within a relatively narrow context. The evaluation metrics focusing on alignment rates with human judgment are reasonable, though the reliance on just three primary human evaluators introduces potential limitations to the robustness of the ground truth establishment.

**Other Comments Or Suggestions:**

See above.

**Other Strengths And Weaknesses:**

See above.

**Questions For Authors:**

1: How generalizable is the Agent-as-a-Judge framework beyond code generation tasks? The current evaluation is limited to DevAI's code generation tasks. Could you elaborate on how the framework would be adapted for substantially different domains (e.g., multimodal reasoning, open-ended dialogue, or physical environment navigation)?

2: The paper shows Agent-as-a-Judge sometimes outperforms individual human evaluators. Given the small evaluator pool, how confident are you in this conclusion? What additional validation would strengthen this claim?

3: DevAI contains 55 tasks, which is relatively small compared to some benchmarks. What specific challenges prevented creating a larger benchmark, and how might the limitation in scale affect your conclusions about the framework's performance?

4: Did you observe any systematic patterns in the types of requirements or tasks where Agent-as-a-Judge performed poorly relative to human evaluation?

**Relation To Broader Scientific Literature:**

The framework builds on several key research streams in AI evaluation. It extends the LLM-as-a-Judge approach (Zheng et al., 2024) by adding agentic capabilities to enable intermediate feedback throughout the task-solving process. The paper connects to the growing literature on agentic systems evaluation, addressing limitations in traditional benchmark approaches like HumanEval (Chen et al., 2021) and SWE-Bench (Jimenez et al.), which focus primarily on final outcomes. The authors' findings on human evaluator disagreement echo research on annotation inconsistency in ML evaluation (Chen et al., 2024c), while their ensemble approach to human consensus aligns with work on judgment aggregation (Hastie et al., 2009).

**Theoretical Claims:**

The paper does not contain formal mathematical proofs for theoretical claims as it is primarily empirical.

---

> ### Author Rebuttal · Authors · 2025-03-31
>
> **Thank you for your valuable feedback and insightful questions. We greatly appreciate your encouraging words for this paper. We will carefully address each concern in detail below.**
>
> > Q1: How generalizable is the Agent-as-a-Judge framework beyond DevAI's code generation tasks, and how could it be adapted to other domains.
>
> We focused on code generation as it has clear intermediate objectives and is a promising application of agentic systems, but the concept of Agent-as-a-Judge is domain-agnostic. Even the modular implementation of our version of Agent-as-a-Judge (e.g. graph parsing, search, read, etc.) already support multiple data modalities, including code, text, images, and videos. We expect that extending Agent-as-a-Judge to other domains might just require one to potentially recombind our modules (potentially adding more functions their own).
>
> As you'll be aware, the field of agentic systems is moving quite quickly. Indeed, several other papers have already directly implemented agent-as-a-judge for other applications, like mobile apps [1], long documents [2], bioinformatics [3], github issue [4]:
>
> [1] AutoEval: A Practical Framework for Autonomous Evaluation of Mobile Agents
>
> [2] Agent-as-Judge for Factual Summarization of Long Narratives
>
> [3] BioAgents: Democratizing Bioinformatics Analysis with Multi-Agent Systems
>
> [4] Unveiling Pitfalls: Understanding Why AI-driven Code Agents Fail at GitHub Issue Resolution
>
> > Q2+Comments: How confident are you in the claim that Agent-as-a-Judge outperforms human evaluators? What further validation would strengthen this? Also, how do you address inconsistencies in black-box vs. gray-box system comparisons?
>
> In this work, our human baseline was obtained via an ensemble of experts with 2 rounds of processing to ensure high-quality judgments (which Agent-as-a-Judge matched closely). Thus we are fairly confident that our results are accurate and our alignment rates roughtly align with an expected alignment rate.
>
> To address the risk that our small sample size could influence the results, as you noted, we conducted an extended analysis with ten additional evaluators. The purpose of this was to determine if the alignment rate between our primary three evaluators was reliable. Indeed, the results of this validated our belief in that assumption.
>
> The only difference between the black-box and gray-box settings is whether there are trajectory records. We encourage users to provide their trajectories, as SWE-Bench [5] did.
>
> [5] SWE-bench: Can Language Models Resolve Real-World GitHub Issues?
>
> > Q3: Given DevAI's relatively small size (55 tasks), what challenges limited its scale, and how might this affect conclusions about the framework's performance?
>
> In total, our benchmark yields 365 evaluation data points, compensating for the moderate number of tasks by introducing multi-step tasks. We made this design choice deliberately: quality over quantity. By having fewer tasks but making them comprehensive and realistic, we ensure that solving them requires diverse skills (planning, coding, possibly handling multi-modal data, etc.), which better stress-tests agentic systems than simpler tasks would. Creating DevAI did involve substantial manual effort (annotating 365 requirements with ground-truth outcomes), so we limited its size to ensure each task was well-curated and reflective of real-world challenges. Notably, our results show that current state-of-the-art agentic code systems could fully solve only 1 out of 55 tasks, and on average met ~29% of the requirements (when evaluated independently)---indicating these tasks pose a challenge even to advanced agents. Besides, finish the tasks are not super cheap (which may be a burden for ), for examples, OpenHands use $350.9 to complete all the tasks. In short, there is no significant reason for having more tasks at this time.
>
> > Q4: Any patterns where Agent-as-a-Judge lagged behind humans?
>
> Me provide an analysis of the failure cases of Agent-as-a-Judge (relative to the human evaluation) in Appendix N. Agent-as-a-Judge struggled most with data preprocessing/postprocessing tasks, although errors were often human-like. Altogether, this gives a strong path for improving the included implementation of Agent-as-a-Judge, but we leave this as future work.
>
> | Category | Count |
> |---|---|
> | Data preprocessing and postprocessing | 10 |
> | Dataset or Environment | 8 |
> | Other | 5 |
> | Machine Learning Method | 4 |
> | Performance Metrics | 3 |
> | Visualization | 3 |
> | Human-Computer Interaction | 3 |
>
>
>
>
> ---
>
> **We appreciate your careful evaluation and valuable insights again.**

---

### Official Review · Reviewer_yar8 · 2025-03-17

**Overall Recommendation:** 3

**Summary:**

This paper proposes a new evaluation framework which uses agentic systems to judge a subject agentic system. To validate and demonstrate this framework, the paper also introduces a new benchmark of code generation tasks with manual annotations. The authors benchmark several existing code-generating agentic systems with the new benchmark dataset using agent-as-judge and LLM-as-judge frameworks and find that the agent-as-judge framwork out-performs LLM-as-judge.



### Update after rebuttal

The authors have addressed most of my and other reviewers' questions satisfactorily. Therefore, I am upgrading my rating.

**Claims And Evidence:**

Most of the claims are supported by evidence. Please see the strengths and weaknesses section for problematic claims and questions.

**Essential References Not Discussed:**

No

**Experimental Designs Or Analyses:**

Yes. The experimental analyses are sound and valid.

**Methods And Evaluation Criteria:**

Yes

**Other Comments Or Suggestions:**

1. In lines 80-82 (left column), it should be "proving" or "following" instead of "violating".

2. A lot of the actual details about the method and the dataset have been moved to the appendix. A lot of these need to be in the main paper for the review process to be fair. For example, at some of the details about the sample collection and human labeling process mention in Section 2.2 (and given in Appendix E), should be in the main paper.

3. In Section 3.1 (Performance Analysis), the paper mentions some "intermediate requirements". What are these? These have not been mentioned in anywhere else in the paper.

**Other Strengths And Weaknesses:**

Agentic systems as a judge can be a good direction for this area of research. However, the arguments and evidence presented in this paper raises a lot of questions. I would like the authors to consider these questions in their rebuttal and any revisions.

1. What is stopping DevAI from becoming another measure that becomes a target for the community, thereby making it similar to other benchmarks?

2. The methodology for comparing against human evaluators is not convincing. In lines 106-108, the paper provides savings in terms of cost and time. Are these compared to human evaluators? These time and cost savings are subjective. Human evaluators could be more skilled, thereby requiring less time, or less skilled. In that case, you might want to pay  more or less, thereby changing the cost savings also. So, the evidence does not seem very strong.

3. Section 2.2 talks about "softer requirements". How are these weighted vs the hard requirements? Is there explicit weighting somewhere or just asking the evaluation agent?

4. Section 3.1 gives the time spent by humans on evaluations and discussions. How were these hours spread over days? Using only 3 evaluators has a chance of introducing biases - were the humans tired after long hours?, would they give the same scores on another day? Usually, it's better to use a larger set of human evaluators to reduce the average biases.

5. Lines 203-204 mention that the OpenHands processed at most 1252482 tokens. What does "processing" tokens mean here? If I understand correctly, the same number of input tokens are give to all models - are these the intermediate "thought" or "action" tokens generated?

6. In Section 3.1 (Performance Analysis), what are pre-requisites? What does "ignoring" mean? Does it mean that the prerequisites are assumed to be satisfied, or does it mean that you don't care whether they are satisfied?

7. For lines 211-212 - are these the average number of lines in the saved code files? If yes, does MetaGPT really only generate 11.15 lines of code and saves only 0.42 files - for an average of only <5 lines of code?

8. In Section 3.2 (Disagreement Analysis), what are the evaluators disagreeing on? Are they disagreeing on whether the code satisfies the requirements or are they disagreeing on whether the code achieves the goal or are they disagreeing on whether the code is efficient?

9. Lines 285-290 talk about the performance of evaluator cn9o. The low performance clearly means that this evaluator is not at a very high quality. So, the bias in evaluation mentioned above will be even higher. How do the authors account for such issues, particularly when using such few evaluators?

10. In Section 4.3, would the order in which these components are added impact the performance? Or to put it another way - are there second order dependencies among these components such that some components are only useful in the presence of some other components?

**Questions For Authors:**

Please see strengths and weaknesses.

**Relation To Broader Scientific Literature:**

The paper is related to prior work in model-based evaluations, specifically the literature on LLM-as-a-judge. The paper presents a different framework for evaluating agentic-systems - agent-as-a-judge.

**Theoretical Claims:**

NA

---

> ### Author Rebuttal · Authors · 2025-03-31
>
> **We sincerely appreciate your time and insightful feedback, as well as your acknowledgment of our work’s novelty.**
>
> > Q1. What is stopping DevAI from becoming another measure that becomes a target for the community?
>
> DevAI is specifically designed to evaluate the entire intermediate automated AI development rather than just the final output. Because it analyzes step-by-step reasoning and diverse intermediate results, it is much harder to “game” by tuning solely for a single end metric.
>
> > Q2. The method for comparing against human evaluators is not convincing. L 106-108 provide cost/time savings, but these seem subjective.
>
> We base our numbers on a pilot study. For reference, we used \$15 USD/hour and expert evaluators with over 5 years of AI experience. Human evaluation can vary with expertise, but to create a fair baseline, we:
> 1. Chose a moderate pay rate for calculations.
> 2. Used multiple evaluators in a two-round process to discuss, reconcile, and enhance reliability.
>
> We do not claim these figures are universal, only that they show our automated evaluator can offer considerable time/cost savings compared to human evaluation in many scenarios.
>
> > Q3. Sec 2.2 talks about "softer requirements." How are these weighted versus hard requirements? Is there explicit weighting somewhere or just asking the evaluation agent?
>
> Soft requirements (preference) are part of real-world developer tasks but remain subjective. Our current experiment focuses only on objective requirements. We do not formally evaluate these preferences here and leave a deeper exploration of how the framework might weigh subjective or soft criteria in future work.
>
> > Q4. Sec 3.1 gives the time spent by humans on evaluations. How were these hours spread over days? Using only three evaluators can introduce biases—were they tired? Would they give the same scores another day?
>
> The three evaluators worked over seven days. Spreading it out reduced fatigue and let them pace themselves. We recognize that having only three evaluators can introduce bias or variance, but recruiting many expert reviewers is difficult. To mitigate this, we:
> - Placed no strict time pressure on evaluators.
> - Implemented a two-round process where evaluators discussed and reconciled any differences in their assessments.
>
> > Q5. L203-204 mention that OpenHands processed at most 1,252,482 tokens. What does “processing” tokens mean?
>
> This refers to the total number of tokens a method reads or generates during its entire reasoning process. Although each system sees the same user queries, they differ in how much intermediate reasoning is conducted, leading to varying token counts.
>
> > Q6. In Sec 3.1 (Performance Analysis), what are “prerequisites”? What does “ignoring” them mean?
>
> Some requirements depend on satisfying preliminary sub-requirements first. We use two modes:
> 1. **Ignoring prerequisites:** Each requirement is scored independently, treating each one like a standalone test.
> 2. **Enforcing prerequisites:** If a prerequisite is unmet, all subsequent requirements tied to it automatically fail.
>
> > Q7. In L211-212, is 11.15 the average number of lines in saved code files? MetaGPT seems to generate few lines of code.
>
> These numbers reflect MetaGPT’s average output. It often failed to save code, averaging 0.42 files per task with about 11 lines each. This shows that our benchmark captures issues that other benchmarks miss. We reported this to the MetaGPT team, and they have since improved it.
>
> > Q8. In Sec 3.2 (Disagreement Analysis), what exactly did the evaluators disagree on?
>
> They disagreed on whether certain requirements were met. One evaluator might decide a requirement was fulfilled, while another concluded it was partially or completely unmet. Our second-round review helped reconcile these views.
>
> > Q9. L285-290 discuss evaluator cn9o’s low performance. This suggests evaluation bias with only a few evaluators.
>
> We used the second-round debate to address cn9o’s different judgments. Sometimes cn9o caught overlooked issues; other times, cn9o was too strict. The group either aligned with cn9o’s finding or persuaded cn9o to adjust. This consensus-building reduces any single evaluator’s bias.
>
> > Q10. In Sec 4.3, does adding components in different orders affect performance?
>
> Yes. Some components are interdependent (e.g., a file-reading module is not very useful without a file-finding module). This is typical in agentic systems. We conducted ablations and used an intuitive component ordering, which is common practice.
>
> > Other Comments
>
> 1. Thanks, we will fix this.
> 2. Due to strict page limits, some details were placed in the appendix; we will consider shifting more details (e.g., from Appendix E) to the main text.
> 3. "Intermediate requirements" are the same with "requirements" mentioned in the paper.
>
> ---
>
> **Thank you for your insights and time again**

---

### Decision · Program_Chairs · 2025-05-01

**Decision:**

Accept (poster)

**Comment:**

All reviewers agree that this paper is well-written and the solution is practical and effective. The proposed Agent-as-a-Judge framework can be potentially applied in various tasks and settings. We hope the additional discussion and results in the rebuttal can be integrated into the final version of this paper to make it more solid and complete.